



# Sensitivity of spatial aerosol particle distributions to the boundary conditions in the PALM model system 6.0

Mona Kurppa[1], Pontus Roldin[2], Jani Strömberg[1], Anna Balling[1], Sasu Karttunen[1], Heino Kuuluvainen[3], Jarkko V. Niemi[4], Liisa Pirjola[5], Topi Rönkkö[3], Hilkka Timonen[6], Antti Hellsten[6], and Leena Järvi[1,7]

[1]Institute for Atmospheric and Earth System Research, Faculty of Science, University of Helsinki, Helsinki
[2]Division of Nuclear Physics, Lund University, Lund, Sweden
[3]Aerosol Physics Laboratory, Physics Unit, Tampere University, Tampere, Finland
[4]Helsinki Region Environmental Services Authority (HSY), Helsinki, Finland
[5]Department of Automotive and Mechanical Engineering, Metropolia University of Applied Sciences, Vantaa, Finland
[6]Atmospheric Composition Research, Finnish Meteorological Institute, Helsinki, Finland
[7]Helsinki Institute of Sustainability Science, University of Helsinki, Helsinki

**Correspondence:** Mona Kurppa (mona.kurppa@helsinki.fi), Leena Järvi (leena.jarvi@helsinki.fi)

**Abstract.**

High-resolution modelling is needed to understand urban air quality and pollutant dispersion in detail. Recently, the PALM model system 6.0, which is based on the large-eddy simulation (LES), was extended with a detailed aerosol module SALSA2.0 to enable studying the complex interactions between the turbulent flow field and aerosol dynamic processes. This study rep-

5 resents an extensive evaluation of the modelling system against the horizontal and vertical distributions of aerosol particles measured using a mobile laboratory and a drone in an urban neighbourhood in Helsinki, Finland. Specific emphasis is on the model sensitivity of aerosol particle concentrations, size distributions and chemical compositions to boundary conditions of meteorological variables and aerosol background concentrations. The meteorological boundary conditions are drawn from both a numerical weather prediction model and observations, which occasionally differ strongly. Yet, the model shows good

agreement with measurements (fractional bias $< 0.67$, normalised mean-square error $< 6$, factor of two $> 0.3$, normalised mean bias factor $< 0.25$ and normalised mean absolute error factor $< 0.35$) in respect of both horizontal and vertical distribution of aerosol particles, their size distribution and chemical composition. The horizontal distribution is most sensitive to the wind speed and atmospheric stratification and vertical distribution to the wind direction. The aerosol number size distribution is mainly governed by the flow field along the main street with high traffic rates and in its surroundings by the background

concentrations. The results emphasize the importance of correct meteorological and aerosol background boundary conditions, in addition to accurate emission estimates and detailed model physics, in quantitative high-resolution air pollution modelling and future urban LES studies.





# 1   Introduction

Exposure to outdoor air pollution is a major global threat resulting up to 0.8 million premature deaths in Europe (Lelieveld et al., 2019) and 3 million worldwide (Lelieveld et al., 2015; WHO, 2016) every year. Specifically aerosol particles can be extremely harmful, and based on a recent study by Burnett et al. (2018) outdoor fine particulate air pollution ($PM_{2.5}$) solely could have

caused up to 8.9 million deaths worldwide in 2015. As over half of the global population lives in cities (55 % according to UN, 2019), urban air quality is of major importance. In addition to high population densities, urban areas are characterized by major air pollutant sources, namely traffic exhaust and road dust, being at the same level where urban dwellers inhale outdoor air. The dispersion of these traffic-related pollutants is not, however, straightforward as buildings, trees and other obstacles modify the flow within the urban canopy and hence also pollutant dispersion (Tominaga and Stathopoulos, 2013) as well as the

environment for aerosol dynamic processes and chemical reactions to occur.

As a consequence of the complex interactions between the urban morphology, meteorology, local emissions and air pollutant dynamics and chemistry, air quality is highly variable both in time and space, and strong concentration gradients are observed in urban areas. However, measurements from a single monitoring station nearest to the individual's residence, hospital, or primary health care clinic have commonly been applied in air pollution exposure studies (Andersen et al., 2012; Adam et al.,

2015), which can lead to notable errors. Moreover, both the size and chemical composition of aerosol particles are of major importance when it comes to their health impacts (Kampa and Castanas, 2008; Kelly and Fussell, 2012). For instance, particle deposition in lungs depends strongly on the inhaled particle size (Hussain et al., 2011), and thus the negative health effects of aerosol particles have been found to correlate more strongly with the surface area of particles than their number or mass (Brown et al., 2001; Oberdörster et al., 2005).

Computational fluid dynamics (CFD) models have been successfully applied in studying the air flow and dispersion of air pollutants in urban areas. Mainly models based on either Reynolds-averaged Navier Stokes (RANS, e.g., Baik et al., 2009; Kwak et al., 2015; Santiago et al., 2020) or large-eddy simulation (LES, e.g., García-Sánchez et al., 2018; Letzel et al., 2012; Salim et al., 2011) have been utilised. While being computationally more expensive than RANS, LES has been shown to perform better in resolving instantaneous turbulence structures in a complex urban environment (García-Sánchez et al., 2018;

Salim et al., 2011). Further, air pollutant concentrations can be significantly modified by their chemical and physical processes (Kurppa et al., 2019; Nikolova et al., 2016; Zhong et al., 2020), especially as the residence time of air pollutants is increased in a complex urban environment (Gronemeier and Sühring, 2019; Ramponi et al., 2015). Therefore a detailed module describing the characteristics of air pollutants and their dynamics is needed to enable modelling aerosol particles of different size, chemical composition and harmfulness. To date, only a few LES models include a module for treating aerosol particles with a specific

size distribution and chemical composition and their dynamic processes (Kurppa et al., 2019; Steffens et al., 2013; Zhong et al., 2020).

The sectional aerosol module SALSA (Kokkola et al., 2008) was recently implemented to the PALM model system (Kurppa et al., 2019) to consider the impact of aerosol dynamic processes on aerosol concentrations and size distributions, and to study the relative importance of pollutant dispersion and aerosol dynamic processes. A model evaluation in central Cambridge, UK,



showed the model to be capable of reproducing the vertical distributions of aerosol size distribution in a simple street canyon. However, due to the lack of observations, the capability of the model to reproduce the horizontal distributions of aerosol particles has not been studied yet. Also the meteorological conditions were limited to one single day and the examined street canyon had no vegetation.

Still, even if the air pollutant processes would be modelled accurately, correct boundary conditions for the meteorological variables and air pollutant concentrations are vital for realistic air quality simulations. Boundary conditions can be drawn from observations, which however are typically point measurements that lack spatial representatives and also are prone to measurement errors. Another alternative is to use model data, which provide a good spatial coverage but not necessarily stable performance. Previously, CFD models have been successfully coupled with mesoscale models to study the impact

of larger scale atmospheric features on microscale interactions (e.g., Baik et al., 2009; Heinze et al., 2017; Liu et al., 2012; Michioka et al., 2013; Wyszogrodzki et al., 2012) as well as to consider realistic air pollutant background concentrations (Kwak et al., 2015). Recently, Santiago et al. (2020) investigated the sensitivity of RANS-based urban $PM_{10}$ (particulate matter with aerodynamic diameter $< 10\,\mu m$) simulations on the meteorological boundary conditions and showed the model performance to be improved when replacing the wind direction (WD) predicted by the WRF model with the observed WD. However, Santiago

et al. (2020) only modelled passive $PM_{10}$ without taking into account chemical or physical transformation of aerosol particles. Hence, it is still unclear how much uncertainties in aerosol particle concentrations and size distributions can model boundary conditions cause.

To further assess the performance of SALSA2.0 in the PALM model system 6.0 in simulating the spatial distribution of aerosol particle concentrations in an urban area and to examine the importance of meteorological and aerosol background

boundary conditions, we will use observations made during an extensive measurement campaign in an urban neighbourhood in Helsinki, Finland, in summer and winter 2017. The campaign focused on the spatial variability of aerosol particle number, surface area and mass both in horizontal and vertical as well as aerosol size distributions and chemical composition with a high temporal and spatial resolution measured using a mobile laboratory and a drone. The model evaluation is done at three observation periods with different prevailing meteorological conditions.

**2 Measurements**

**2.1 Measurement campaign**

The model evaluation and sensitivity study is conducted around an Helsinki Region Environmental Services Authority (HSY) air quality monitoring site, hereafter referred as the "supersite", in Helsinki, Finland (60°11'47"N, 24°57'07"E). The site is located 3 km north-northeast from the Helsinki city centre, and it is characterised as an urban street-canyon kerbside station

with a traffic rate of around 28,000 on a workday, of which 12 % are heavy duty vehicles (City of Helsinki, 2018). The street canyon is 42 m wide and the mean building height is around 19 m on the southwestern and 16 m on the northeastern side of the street (see Fig. 1 in Kuuluvainen et al., 2018) resulting in a height to width ratio of 0.42. The supersite consists of a container



**Table 1.** Instrumentation of the mobile laboratory Sniffer. Abbreviations: PSD = aerosol particle number size distribution, $N_{\text{tot}}$ = total aerosol particle number concentration and $PM_1$ = mass of particulate matter with aerodynamic diameter $< 1$ µm.

| Measured component | Instrument |
| --- | --- |
| PSD (5.6–560 nm) | Engine exhaust particle sizer (EEPS, model 3090, TSI) |
| PSD (7 nm–10 µm) | Electrical low-pressure impactor (ELPI, Dekati Ltd) |
| $N_{\text{tot}}$ ($> 2.5$ nm) | Butanol condensation particle counter (CPC, model 3776, TSI) |
| Black carbon (in $PM_1$) | Aethalometer (Model AE33, Magee Scientific) |

(length 8.0 m, width 1.7 m and height 2.7 m) equipped with standard air quality measurement devices measuring from 4 m above ground level.

To get information about the spatial variability of air pollutants around the supersite, a two-week measurement campaign was conducted in summer (6–16 Jun) and winter (28 Nov–11 Dec) 2017. During both campaigns the horizontal distribution of

air pollutants in the neighbourhood was monitored on non-rainy days using a mobile laboratory and additionally during two intensive observation periods the vertical profiles of aerosol particles were measured using a drone.

The mobile laboratory Sniffer (Pirjola et al., 2004) measured the horizontal distribution of trace gases and aerosol particle concentrations and size distribution. The measurements were done in one to two hour slots with a 1-s temporal resolution during the morning and afternoon rush hours, around noon and in the late evening. During each observation period, Sniffer

was driving along a main street (Mäkelänkatu) and a side street as well as standing at the supersite, opposite the supersite and on a field 185 m from the main street (hereafter "background"). The instrumentation of Sniffer is given in Table 1 and the measurement locations in Fig. S1 in the Supplement. The main inlet was situated above the windshield at 2.4 m and a global positioning system (model GPS V, Garmin) recorded the van speed and position. For a detailed description on Sniffer, see Enroth et al. (2016); Pirjola et al. (2016).

During the intensive observation periods, a multi-rotor drone (X8, VideoDrone Finland Ltd) carried an electrical particle sensor (Partector, Naneos GmbH) to measure the vertical distribution of the alveolar lung-deposited surface area (LDSA) of aerosol particles, which describes the total aerosol surface area penetrating to the deepest parts of lungs (see e.g., Kuula et al., 2020, and references within). The measurement were done on both sides of the street canyon when the Sniffer was simultaneously driving. The drone was flown ten times up-and-down between $z = 2 - 50$ m during one 30-minute measurement

interval, after which measurements were repeated on the other side. Each intensive observation period started by measuring LDSA at the supersite and ended on the other side. Measurements were started at 3 m from the building wall and the horizontal location was kept constant with a GPS sensor of the drone. Additionally, LDSA was measured at the supersite by a Pegasor AQ Urban sensor (Pegasor Ltd.) and on the other side by a DiSCmini (Testo Ltd.) or with another Partector at 1 m and in winter also at 14 m. For the details of the instrumentation, see Kuuluvainen et al. (2018).

The sensitivity of the results to the PALM model boundary conditions is examined during the following three periods: 9 Jun morning (07:16–09:15) and evening (20:26–21:14), and 12 Dec morning (07:20–09:14).



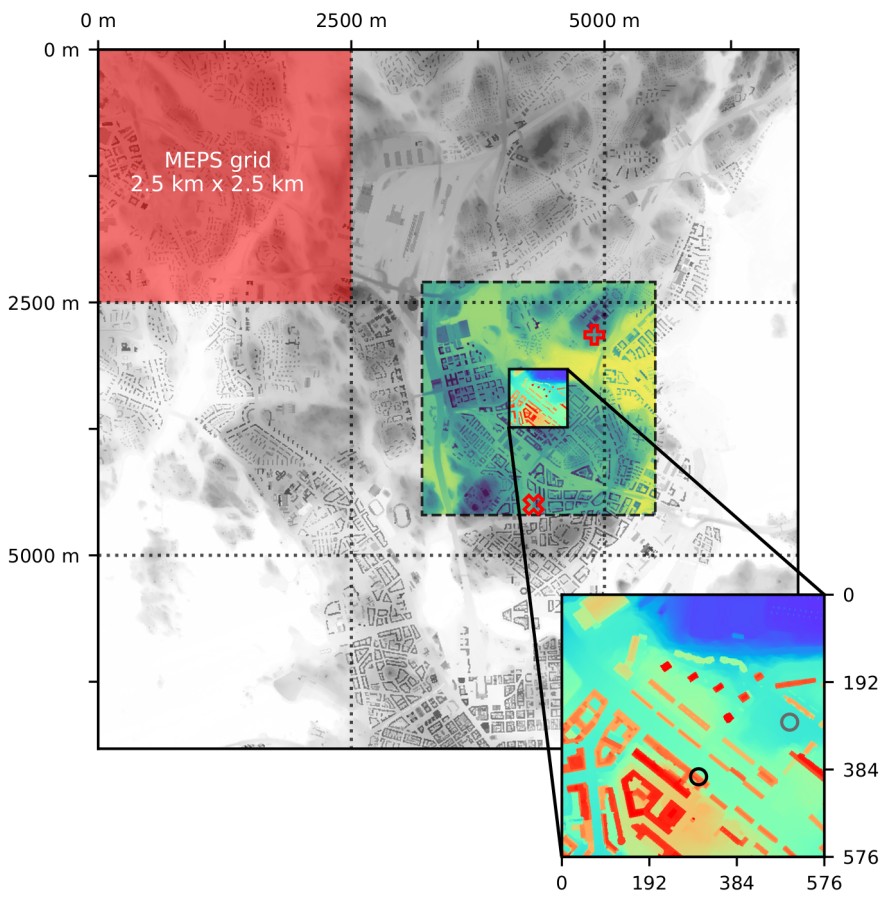

**Figure 1.** The modelling domain: root in grey, parent in viridis (dashed line) and child in rainbow (solid line). The grid size of the MEPS data is illustrated with a red area and dotted lines. The background air quality monitoring sites are marked with a red empty plus (SMEAR III) and a red empty cross (Kallio). In the zoomed figure over the child domain, the supersite is marked with a black circle and the background measurement point of Sniffer with a grey circle.

## 2.2 Additional measurements

In addition to the Sniffer and drone measurements, we use stationary aerosol observations from the supersite and two urban background monitoring sites: Kallio site operated by HSY and SMEAR III (Station for Measuring Ecosystem Atmospheric relations, Järvi et al., 2009) around $1.0$ km southwest and $0.8$ km northeast from the supersite, respectively (Fig. 1). See Table S1 for the instrumentation. In addition to aerosol observations, meteorological data (wind speed, wind direction, air temperature) from the SMEAR III measurement tower ($z = 31$ m) and Kivenlahti meteorological measurement mast $17.4$ km west from the supersite (Wood et al., 2013) are used in the study.





## 3 Simulations

### 3.1 Model description

This study applies the PALM model system, version 6.0 (revision 4416) (Maronga et al., 2015, 2020), which features an LES core for atmospheric and oceanic boundary layer flows. PALM solves the non-hydrostatic, filtered, incompressible Navier-Stokes equations of wind ($u$, $v$, and $w$) and scalar variables (sub-grid-scale turbulent kinetic energy $e$, potential temperature $\theta$, and specific humidity $q$) in Boussinesq-approximated form. PALM is especially suitable for complex urban areas, owing to its features such as a Cartesian topography scheme and a plant canopy module, which are applied here to include the aerodynamic impact of both solid buildings and permeable vegetation on the flow. Furthermore, so called PALM-4U (short for PALM for urban applications) components have recently been implemented to PALM (Maronga et al., 2020), including the aerosol module SALSA, the online chemistry module, and the self- and offline nesting features, which are all applied in this study.

SALSA (Kokkola et al., 2008; Kurppa et al., 2019) describes an aerosol size distribution by a number of size bins (10 by default) and each bin can be composed of different chemical components. Chemical components included are sulfuric acid ($H_2SO_4$), organic carbon (OC), black carbon (BC), nitric acid ($HNO_3$), ammonium ($NH_3$), sea salt, dust, and water ($H_2O$). SALSA contains the following aerosol dynamic processes: coagulation, nucleation, dry deposition on solid surfaces and resolved-scale vegetation, and condensation and dissolutional growth by gaseous $H_2SO_4$, $HNO_3$, $NH_3$, and semi- and non-volatile organics (SVOC and NVOC). The gaseous compounds can be transferred to SALSA from the online chemistry module, which is based on Kinetic Pre-Processor (KPP, Damian et al., 2002) version 2.2.3 and an adapted version of the KP[4] pre-processing tool (Jöckel et al., 2010). The implementation is flexible, allowing the user to choose the chemical mechanism and components being considered. In this study, a simplified mechanism describing photochemical smog is applied (see Supplement, Section S3.2). Photolysis is parametrised based on Saunders et al. (2003).

To capture the dominant turbulent eddies of the atmospheric boundary layer (ABL) in LES, the horizontal extent of the modelling domain should span over several ABL heights, see e.g. (Fishpool et al., 2009; Chung and McKeon, 2010; Auvinen et al., 2020). At the same, to resolve most of the kinetic energy within street canyons, a high enough grid resolution is needed (Xie and Castro, 2006). Furthermore, uncertainty arising from the lateral boundary conditions usually decreases with increasing horizontal dimensions. To fulfill these contradicting requirements, a self-nesting feature has been included in PALM (Hellsten et al., 2017; Maronga et al., 2020). In self-nesting, one or several child domains are nested within a parent domain and the child obtains its boundary conditions from its parent. Furthermore, PALM incorporates an automated mesoscale offline nesting with a mesoscale operational weather prediction model, which allows realistic, non-cyclic and non-stationary boundary conditions for the flow. To reduce the time and distance for the mesoscale flow field to adjust to the LES modelling domain, a synthetic turbulence generator within PALM can be applied.

### 3.2 Model domain and morphological data

The model simulations are conducted over a root domain of 6.9 km × 6.9 km, within which two smaller domains, parent and child, are nested progressively (Fig. 1). The dimensions ($L_x$, $L_y$, $L_z$), number of grid points ($N_x$, $N_y$, $N_z$) and grid resolutions





**Table 2.** Dimensions ($L$), number of grid points ($N$) and grid resolutions ($\Delta$) of the model domains in $x$-, $y$- and $z$-directions.

| Domain | $L_x \times L_y \times L_z$ (m$^3$) | $N_x \times N_y \times N_z$ | $\Delta_x, \Delta_y, \Delta_z$ (m) |
|---|---|---|---|
| Root | $6{,}912 \times 6{,}912 \times 606$ | $768 \times 768 \times 80$ | $9.0, 9.0, 6.0^*$ |
| Parent | $2{,}304 \times 2{,}304 \times 288$ | $768 \times 768 \times 96$ | $3.0, 3.0, 3.0$ |
| Child | $576 \times 576 \times 144$ | $576 \times 576 \times 144$ | $1.0, 1.0, 1.0$ |

*: $\Delta_z$ is stretched with a a factor 1.03 above $z = 300$ m resulting in a total domain height of 591 m.

($\Delta_x$, $\Delta_y$, $\Delta_z$) of each domain are given in Table 2. In this study, the focus is on the child domain which matches with the area of the spatial aerosol measurements around the supersite.

Information on the building and vegetation height and land surface elevation are drawn from high-resolution raster maps for Helsinki (Auvinen and Aarnio, 2019). The manipulation of the domain input files is done using the Python library P4UL (Auvinen and Karttunen, 2019). Only vegetation higher than $z_{\mathrm{v,min}} = 4.0$ m are included in the simulations. Due to the lack of observational data on the leaf area density (LAD) of vegetation, a constant LAD value is applied for all tree crowns above $z_{\mathrm{v,min}}$. In summer, LAD $= 1.2$ m$^2$ m$^{-3}$ for broad-leaf trees (Abhijith et al., 2017), while in winter LAD is decreased to 20 % of the summertime value.

### 3.3 Meteorological boundary conditions

We apply both modelled and observed data as meteorological boundary conditions, which are set dynamic, i.e., they change with time.

As modelled data, numerical weather prediction data from MetCoOp Ensemble Prediction System (MEPS, Bengtsson et al., 2017; Müller et al., 2017) are applied. MEPS data were downloaded from the data archive (Norwegian Meteorological Institute, b) using the File Interpolation, Manipulation and EXtraction (Fimex) library (Norwegian Meteorological Institute, a). MEPS has a horizontal resolution of 2.5 km (see Fig. 1), 65 vertical levels and ten ensemble members. It is ran four times daily with a three-hourly cycling for data assimilation (3D-VAR). The lateral boundary data are from the European Centre of Medium-Range Forecasts (ECMWF) high resolution (HRES) atmospheric model. In this study, the MEPS control run, i.e., the ensemble member 0 with unperturbed initial and lateral boundary conditions, is used.

Data from the Kivenlahti mast are downloaded from FMI Open Data service (Finnish Meteorological Institute) as 10-minute-averaged. On the mast, meteorological observations are conducted at three to eight measurement levels between $z = 2 - 327$ m. Despite being located 17.4 km from the supersite, the closest observations of the vertical profile of basic meteorological variables are conducted at Kivenlahti.

The initial conditions and dynamic meteorological boundary data are provided to PALM in a so-called dynamic driver. Of the MEPS data, the dynamic driver was created by the following procedure. First, the sigma-coordinates were translated to pressure coordinates and further to height coordinates applying the hypsometric equation. Then $u$, $v$, $w$, $\theta$ and water vapour





mixing ratio $q_v$ were interpolated from the MEPS grid to the PALM grid: first in horizontal over a two-dimensional grid using the cubic spline method and then in vertical using the linear interpolation. The Kivenlahti mast observations, instead, were linearly interpolated in vertical until the highest observation level, after which a constant value was used. The dynamic driver created from the observational data does not include any horizontal variation.

A mesoscale interface, INIFOR, has been developed to transform mesoscale modelling data into PALM-readable boundary data. However, it is currently only available for COSMO-DE/D2 datasets, which do not cover Finland.

### 3.4    Air pollutant background concentrations

Similar to the meteorological boundary conditions (Section 3.3), both modelled and observed air quality data are used as background concentrations in the simulations. As in the previous model evaluation study (Kurppa et al., 2019), the modelled
background aerosol particle number and trace gas concentrations are produced with the trajectory model for Aerosol Dynamics, gas and particle phase CHEMistry and radiative transfer (ADCHEM, Roldin et al., 2011b, a, 2019). ADCHEM is operated as a one-dimensional column trajectory model along HYSPLIT (Stein et al., 2015) air mass trajectories, starting seven days backwards in time (see Fig. S2 and S3 in the Supplement). The gas and aerosol particle compositions and size distributions are simulated along the back trajectories arriving to the coordinates of the supersite. For the emission inventories and parametri-
sations applied, see Section S3.4 in the Supplement. Detailed descriptions of the aerosol and cloud microphysics, new particle formation and gas-phase chemistry mechanisms in ADCHEM are provided by Roldin et al. (2019) and references therein.

To investigate the impact of the background aerosol size distribution (PSD) and concentration on the model simulations, PSD measurements from SMEAR III (see Section 2) are applied as an alternative for the modelled values. For simplicity, ADCHEM data are always used for the chemical composition of aerosol particles and gaseous concentrations.

For each PALM simulation, the concentrations are averaged over the simulation time and these temporally constant vertical profiles are then introduced to the simulation domain by a decycling method, in which background concentrations are fixed at the lateral boundaries.

### 3.5    Air pollutant emissions

In this study, air pollutant emissions only from traffic combustion are included, as traffic is the main pollutant source within the
modelling domain (Helsinki Region Environmental Services Authority). Traffic-lane maps separating different road categories, i.e., main streets, collector roads and residential streets, have been generated by combining lane and street type information from the Map Service (City of Helsinki). The lane width is 3.5 m. Emissions are introduced as dynamic surface fluxes.

Aerosol particle emission inventories are typically provided as total mass emissions $\text{EF}_{\text{PM}_{2.5}}$. In SALSA, these would need to be translated to number emissions $\text{EF}_N$, assuming some size distribution for the emitted aerosol particles. However,
converting aerosol mass to number is highly sensitivity to the assumed size distribution. Therefore in this study we choose to apply a number emission factor $\text{EF}_N = 4.22 \times 10^{15} \text{ kg}_{\text{fuel}}^{-1}$ based on fuel consumption and a number size distribution estimated by Hietikko et al. (2018) at the supersite in May 2017.





**Table 3.** Unit emission factors for traffic combustion (s = solid and g = gaseous) on 9 Jun between 7:00–8:00 in units $\times 10^{-5}$ $\mathrm{g\,m^{-1}}$ vehicle$^{-1}$. Abbreviations: BC = black carbon, OC = organic carbon, NO = nitrous oxide, NO$_2$ = nitrous dioxide, OCSV = semi-volatile organic carbon, RH = alkanes, H$_2$SO$_4$ = sulphuric acid, N$_2$O = nitrous oxide and NH$_3$ = ammonia.

| BC(s) | OC(s) | NO(g) | NO$_2$(g) | OCSV(g) | RH(g) | H$_2$SO$_4$(g) | N$_2$O(g) | NH$_3$(g) | Fuel |
|-------|-------|-------|-----------|---------|-------|----------------|-----------|-----------|------|
| 1.0   | 0.3   | 49.4  | 13.9      | 0.039   | 1.5   | 0.01           | 1.0       | 3.5       | $9.8 \times 10^3$ |

For gaseous compounds, mass composition of aerosol particles and fuel, unit emission factors $\mathrm{EF_{[compound]}}$ (Table 3) are calculated using emission inventory by the European Environmental Agency for 2017 (Ntziachristos et al., 2016) and specifically the Tier 3 method, which applies information on the mileage per vehicle category and technology, and driving speed. However, since no information on the cumulative mileage for different Euro classes was available, $\mathrm{EF_{NH_3(g)}}$ and $\mathrm{EF_{N_2O(g)}}$ are based on the Tier 1 method (see Ntziachristos et al., 2016, Eq. 28). Furthermore, the following estimates were applied: $\mathrm{EF_{SVOC(g)}} = 0.01 \mathrm{EF_{NMOG}}$ (Zhao et al., 2017, Fig. 2), $\mathrm{EF_{RH(g)}} = 0.4 \mathrm{EF_{NMOG}}$ (Huang et al., 2015, Fig. 4), where NMOG stands for non-methane organic gases and RH(g) for alkanes, and $\mathrm{EF_{H_2SO_4(g)}} = 0.1$ $\mathrm{EF_{SO_2}}$ (Arnold et al., 2006, 2012; Miyakawa et al., 2007). Emitted aerosol particles smaller than 15 nm in diameter are assumed to be composed of 75 % OC and 25 % H$_2$SO$_4$, whereas larger particles contain 72 % BC, 21 % OC and 7 % H$_2$SO$_4$ as estimated from $\mathrm{EF_{PM}}$, $\mathrm{EF_{BC}}$ and $\mathrm{EF_{OC}}$.

The hourly vehicle fleet compositions for the neighbourhood are obtained from the Helsinki Region Environmental Services Authority (HSY and Urban Environment Division of the City of Helsinki, personal communications, 1 Oct, 2018), the mileage for each vehicle technology from the ALIISA model (VTT, 2018) and the fuel sulphur content from the LIPASTO database (VTT, 2017). The traffic rates in the neighborhood are estimated by normalising the mean traffic volumes per each street (Urban Environment Division of the City of Helsinki, Helsinki Region Environmental Services Authority and Helsinki Region Municipalities, 2018) with traffic counts from an online traffic monitoring station located in the northwestern corner of the child domain (City of Helsinki, personal communications, 3 Mar, 2018). Traffic volumes for both southward and northward traffic are measured separately.

## 3.6 Model set-up

The length of the morning simulations on 9 Jun and 12 Dec are two hours, and evening simulation on 9 Jun only one hour. Simulation times correspond to the observation periods.

For all simulation times, two simulations using either modelled (M) or observed (O) boundary conditions for the flow and background aerosol particle number size distribution (PSD) are conducted. The first set-up, hereafter $\mathrm{M_{MET}M_{PSD}}$, applies the modelled meteorological (MET) boundary conditions from the MEPS data and the modelled PSD from the ADCHEM model. The second set-up, hereafter $\mathrm{O_{MET}O_{PSD}}$, applies the observed meteorological data from the Kivenlahti mast and the observed PSD from SMEAR III. Furthermore, two types of sensitivity tests are conducted for the summer morning. Firstly, model sen-





**Table 4.** Simulation abbreviations. WD = wind direction and PSD = aerosol particle number size distribution.

| Simulation | Background meteorology | Background PSD |
|---|---|---|
| $M_{MET}M_{PSD}$ | Modelled by MEPS | Modelled by ADCHEM |
| $O_{MET}O_{PSD}$ | Observed at Kivenlahti | Observed at SMEAR III |
| $M_{MET}O_{PSD}$ | Modelled by MEPS | Observed at SMEAR III |
| $O_{WD,mast}O_{PSD}$ | Modelled, but WD from Kivenlahti | Observed at SMEAR III |
| $O_{WD,SMEAR}O_{PSD}$ | Modelled, but WD from SMEAR III | Observed at SMEAR III |

sitivity on the background PSD is studied running a simulation with the modelled meteorological boundary conditions and observed background PSD ($M_{MET}O_{PSD}$). Secondly, the influence of wind direction (WD) on pollutant dispersion is investigated by replacing WD in the MEPS data by WD measured on the Kivenlahti mast ($O_{WD,mast}O_{PSD}$) or at the SMEAR III station ($O_{WD,SMEAR}O_{PSD}$). As $WD_{SMEAR III}$ is only measured at $z = 31$ m, the wind direction at the model boundaries in

$O_{WD,SMEAR}O_{PSD}$ is set constant with height. In total nine different simulations have been conducted.

The aerosol and chemistry modules are run only within the child domain to limit computational costs. In all simulations, the aerosol processes of condensation and dissolutional growth, coagulation, dry deposition and sedimentation are included and calculated every 1.0 s. The aerosol particle size distribution is described by 10 size bins, of which three are within the first subrange between 2.5–15 nm and seven within the second subrange 15 nm–1 μm. Aerosol particles are assumed to be internally

mixed and hygroscopic, and can contain $H_2SO_4$, OC, BC, $HNO_3$, and/or $NH_3$. The chemical reactions are calculated at every time step of the PALM model.

The advection of both momentum variables and scalars is based on the fifth-order advection scheme by Wicker and Ska-marock (2002) together with a third-order Runge-Kutta time-stepping scheme (Williamson, 1980). The pressure term in the prognostic equations for momentum is calculated using the iterative multigrid scheme (Hackbusch, 1985). The roughness

height is $z_0 = 0.05$ m (Letzel et al., 2012) and the drag coefficient applied for the trees $C_D = 0.3$.

Simulations were first run only for the root domain for 1 h, called here the precursor run, after which the final simulations were started. The final simulations including also the nested parent and child domains were initialised using the final state of the precursor run. Offline nesting is used as forcing for the root domain and the parent and child are nested within using one-way self nesting. As SALSA and chemistry are run only within the child domain, for them the nesting is not applied and

the boundary conditions of air pollutants are set at the child boundaries. The data output was collected starting after the first 15 minutes of the final simulation. Simulations were performed on the Centre for Scientific Computing (CSC) Puhti supercluster. Using in total 394 Intel Xeon processor cores, each simulation required 39–80 h of computing time.





## 4  Results

Section 4.1 describes the differences in the modelled and observed boundary conditions. After that, Section 4.2 focuses on the performance of the simulations applying only modelled ($M_{MET}M_{PSD}$) or only observed ($O_{MET}O_{PSD}$) boundary conditions. Finally, Section 4.3 investigates the impact of background PSD and wind direction on the model performance.

### 4.1  Modelled and observed boundary conditions

The summer morning on 9 Jun is characterized by very calm northerly-north-westerly winds with the horizontal wind speed $U \approx 1.0 - 1.5 \text{ m s}^{-1}$ at $z = 30$ m and mainly $U < 2.5 \text{ m s}^{-1}$ within the lowest 200 m on the Kivenlahti mast (Fig. 2a-b). The MEPS data show more westerly winds, with $-90° < \Delta\text{WD} < -45°$ compared to the Kivenlahti observations, except at 9 am when the modelled and observed WD agree. Furthermore, the observed $U$ are up to $0.5 \text{ m s}^{-1}$ lower within the lowest

100 m during the first two hours and up to $1.0 \text{ m s}^{-1}$ higher during the last two hours when compared to the MEPS data. As the highest measurement level for $U$ on the Kivenlahti mast is $z = 217$ m, the interpolated profile used as the boundary condition in $O_{MET}O_{PSD}$ underestimates $U$ above 217 m at 7 am. The observed and modelled profiles of air ($T$) and dew-point temperature ($T_D$) correspond qualitatively well (Figs. 2c and S4 in the Supplement), but the observations show higher values of $T$ and $T_D$ than MEPS above $z = 200$ m, especially at 8–9 am. Hence, MEPS predicts a stronger and shallower

temperature inversion, which would lead to weaker vertical mixing. Observations at SMEAR III generally follow those on the Kivenlahti mast, expect that $T$ is roughly 2 °C higher at SMEAR III compared to the Kivenlahti mast and WD typically falls between the MEPS data and Kivenlahti observations. The difference in WD can be explained by flow distortion at SMEAR III due to the adjacent buildings to the north of the measurement site (Nordbo et al., 2012). The observed background aerosol particle number concentrations at SMEAR III are around 80 % lower and the modelled PSD shows a smaller peak diameter of

$D = 28$ nm instead of $D = 50$ nm in the SMEAR III observations (Fig. S5). Furthermore, the observations show a secondary peak at $D = 14$ nm, which is not captured by ADCHEM.

By the evening, the observed $U$ on the Kivenlahti mast had increased to $2.0 - 2.5 \text{ m s}^{-1}$ at $z = 30$ m (Fig. S6a) and the wind turned to south-west (Fig. S6b). The modelled and observed WD agree well ($\Delta\text{WD} < 20°$), whereas clear discrepancy is shown for $U$. The MEPS predicts a low-level jet with the maximum $U$ at $z = 100$ m and shows up to $3 \text{ m s}^{-1}$ higher

values compared to the Kivenlahti observations at 8–9 pm. This low-level jet results in a strong wind shear and mechanical turbulence production. Instead above, $U$ is overestimated in the interpolated Kivenlahti data at 9–10 pm. The profiles of $T_D$ agree relatively well (Fig. S7), whereas MEPS predicts clearly lower $T$, with a difference up to $-5$ °C close to the ground (Fig. S6c). The SMEAR III observations agree with those from the Kivenlahti mast. The modelled and observed background PSD agree in shape, but the peak is observed at $D = 79$ nm compared to $D = 100$ nm in ADCHEM and observed total number

concentration is around 35 % lower (Fig. S5).

In the winter morning on 7 Dec, easterly flow was observed and the wind was turning to south-east with both height and time (Fig. S8a-b). Winds were stronger than in the summer morning, around $2 \text{ m s}^{-1}$ at $z = 30$ m. The observed and modelled WD agree, but MEPS predicts up to $3 \text{ m s}^{-1}$ lower $U$ above the canopy. An inversion layer above ground is captured both in



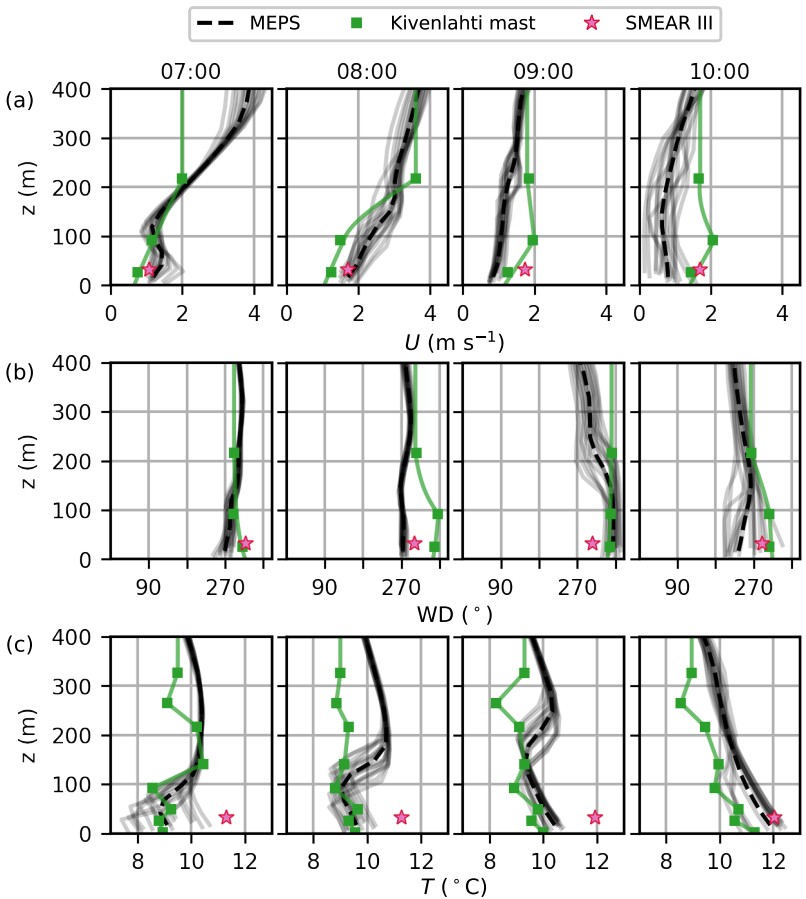

**Figure 2.** Horizontal a) wind speed $U$ (m s$^{-1}$, b) wind direction WD (°) and c) air temperature $T$ °C) on 9 Jun at local time (UTC+3). The modelled profiles at each MEPS grid point are shown by grey solid lines and their mean by a black dashed line. The observation from the Kivenlahti mast are shown by green solid squares and the interpolated profiles used as boundary conditions by a green solid line. Stars show the SMEAR III observations.

MEPS and observations (Fig. S8c), yet it is stronger in the observations especially during the first hours. As a contrast to $T$, MEPS predicts down to $-3\,°\mathrm{C}$ lower $T_D$ compared to the Kivenlahti observations at 9 am (Fig. S9). Similar to the summer morning, the observations on the Kivenlahti mast and SMEAR III are in agreement. Both the modelled and observed PSD peak at $D = 31\,\mathrm{nm}$, but the observed total number concentrations are around 60 % higher (Fig. S10).

## 5 4.2 Evaluation of the air quality modelling results

The model is evaluated against observations at three different observations periods, and in both summer and winter morning the evaluation is done separately for both modelling hours. The following performance measures are applied in the evaluation: fractional bias (FB), normalised mean squared error (NMSE), factor of two (FAC2) (Chang and Hanna, 2004), normalised





**Table 5.** The performance measures and acceptance criteria applied in the evaluation: fractional bias (FB), normalised mean squared error (NMSE), factor of two (FAC2), normalised mean bias factor (NMBF) and normalised mean absolute error factor (NMAEF).

| |FB| | NMSE | FAC2 | |NMBF| | NMAEF |
|---|---|---|---|---|
| <0.67[1] | <6[1] | >0.3[1] | <0.25[2] | <0.35[2] |

1: Hanna and Chang (2012), 2: Yu et al. (2006)

mean bias factor (NMBF) and normalised mean absolute error factor (NMAEF) (Yu et al., 2006). See Appendix A for the definitions and Table 5 for the acceptance criteria. In summary, FB and NMBF measure systematic error (i.e., bias), NMSE and NMAEF the total errors and FAC2 the correct scales. Additionally, the statistical significance of the model error (i.e. the absolute difference between the observations and modelled values) is estimated with the Student's t-test for the horizontal
distributions.

### 4.2.1   Horizontal distribution of total aerosol particle number concentration

In order to compare the data, both the mobile Sniffer measurements containing its geographical coordinates and the PALM data output have been horizontally aggregated to a $5\,\mathrm{m} \times 5\,\mathrm{m}$ grid, with a threshold of at least three measurement points per grid to calculate the median value. A comparison between the measured and modelled total aerosol particle number concentration
($N_{\mathrm{tot}}$) values is illustrated in Fig. 3. In general, the model captures the large concentration gradient between the main street (in the middle from northwest to southeast) and the side street on the northeast side of the main street. However, the model overestimates $N_{\mathrm{tot}}$ at the north-western end of the Sniffer route at all simulation times. Along the side street, the modelled $N_{\mathrm{tot}}$ is slightly higher than the observed in $\mathrm{M_{MET}M_{PSD}}$ during the first hour in the summer morning (Fig. 3f) and in $\mathrm{O_{MET}O_{PSD}}$ during the first hour in the winter morning (Fig. 3n).
Fig. 4 shows the performance measures in simulating the horizontal distribution of $N_{\mathrm{tot}}$ for all simulation times and whether the acceptance criteria are fulfilled. In general, NMBF and NMAEF are more strict measures. NMBF mostly exceeds the acceptance criteria showing that the model tends to over- or underestimate the observations by 25 % or more. NMAEF never fulfills the criteria indicating that the absolute gross error between the observed and modelled values is always over 35 % larger than the mean observation. However, the other measures show mostly acceptable model performance. During the first hour in
the summer morning, $\mathrm{M_{MET}M_{PSD}}$ performs better than $\mathrm{O_{MET}O_{PSD}}$ with respect to all performance measures, as $\mathrm{O_{MET}O_{PSD}}$ clearly overestimates $N_{\mathrm{tot}}$ along the main street (Fig. 3k). For the second hour the difference is small. Instead in the summer evening, $\mathrm{O_{MET}O_{PSD}}$ performs slightly better than $\mathrm{M_{MET}M_{PSD}}$ (e.g., FAC2 = 0.75 and FAC2 = 0.67, respectively), but both acquire good performance values and even NMBF is within the acceptance criteria. In the winter morning, $\mathrm{M_{MET}M_{PSD}}$ fulfills the acceptance criteria during the first hour, except for NMAEF, and overperforms $\mathrm{O_{MET}O_{PSD}}$, but during the second hour
the difference is small. Interestingly, FAC2 is higher for $\mathrm{O_{MET}O_{PSD}}$ than $\mathrm{M_{MET}M_{PSD}}$ over the whole simulation. However, it should be noted that for the first hour the differences in the model absolute error between $\mathrm{M_{MET}M_{PSD}}$ and $\mathrm{O_{MET}O_{PSD}}$ are not significant and for the second hour Student's t-test cannot be performed (see Table S2 in the Supplement).



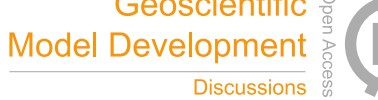

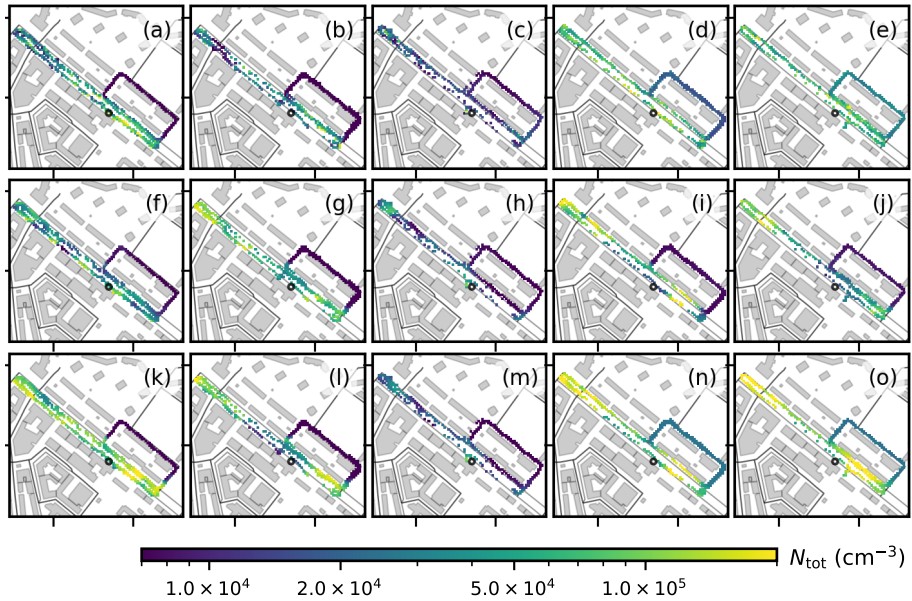

$N_{tot}$ (cm$^{-3}$)

$1.0 \times 10^4$    $2.0 \times 10^4$    $5.0 \times 10^4$    $1.0 \times 10^5$

**Figure 3.** Measured (a-e) and modelled (f-o) total aerosol number concentration ($N_{tot}$) along the Sniffer route for the summer morning (a,f,k for the first and b,g,l for the second hour), summer evening (c,h,m) and winter morning (d,i,n for the first and e,j,o for the second hour). The second row shows $M_{MET}M_{PSD}$ and the third $O_{MET}O_{PSD}$. Measurements are from $z = 2.4$ m and modelled values from $z = 2.5$ m. The supersite is marked with a black circle. Note the different scale in different columns.

### 4.2.2 Vertical profile of the lung-deposited surface area

The modelled vertical profile of alveolar LDSA is evaluated against the observed one over a 5 m × 5 m area next to the supersite on the northern side of the container (Fig. 5) and opposite the supersite on the other side of the main street (Fig. 6). In the summer morning, $M_{MET}M_{PSD}$ performs well opposite the supersite (Figs. 6a-b and 8) especially during the first hour, but it

5   clearly overestimates LDSA at the supersite (Figs. 5a-b and 7). On the contrary, $O_{MET}O_{PSD}$ successfully reproduces the LDSA profile at the supersite (Figs. 5f-g) but not opposite it (Figs. 6f-g). This can be explained by the wind direction: according to the meteorological boundary condition analysis in Section 4.1, the wind direction predicted by MEPS is more westerly than the observed at Kivenlahti. Therefore, a canyon vortex forming in the main street canyon pushes pollutants upwind to the western side of the street in $M_{MET}M_{PSD}$. On the other hand, the opposite is observed for $O_{MET}O_{PSD}$ for which the wind direction is more

10   from the north. In the summer evening, $M_{MET}M_{PSD}$ fulfills all acceptance criteria, while $O_{MET}O_{PSD}$ overestimates LDSA on both sides of the street canyon. This is contradictory to the horizontal distribution of $N_{tot}$, based on which $O_{MET}O_{PSD}$ showed better performance. In the winter morning, $O_{MET}O_{PSD}$ shows slightly better performance at the supersite whereas $M_{MET}M_{PSD}$ underestimates LDSA, while opposite the supersite $O_{MET}O_{PSD}$ clearly overestimates LDSA.





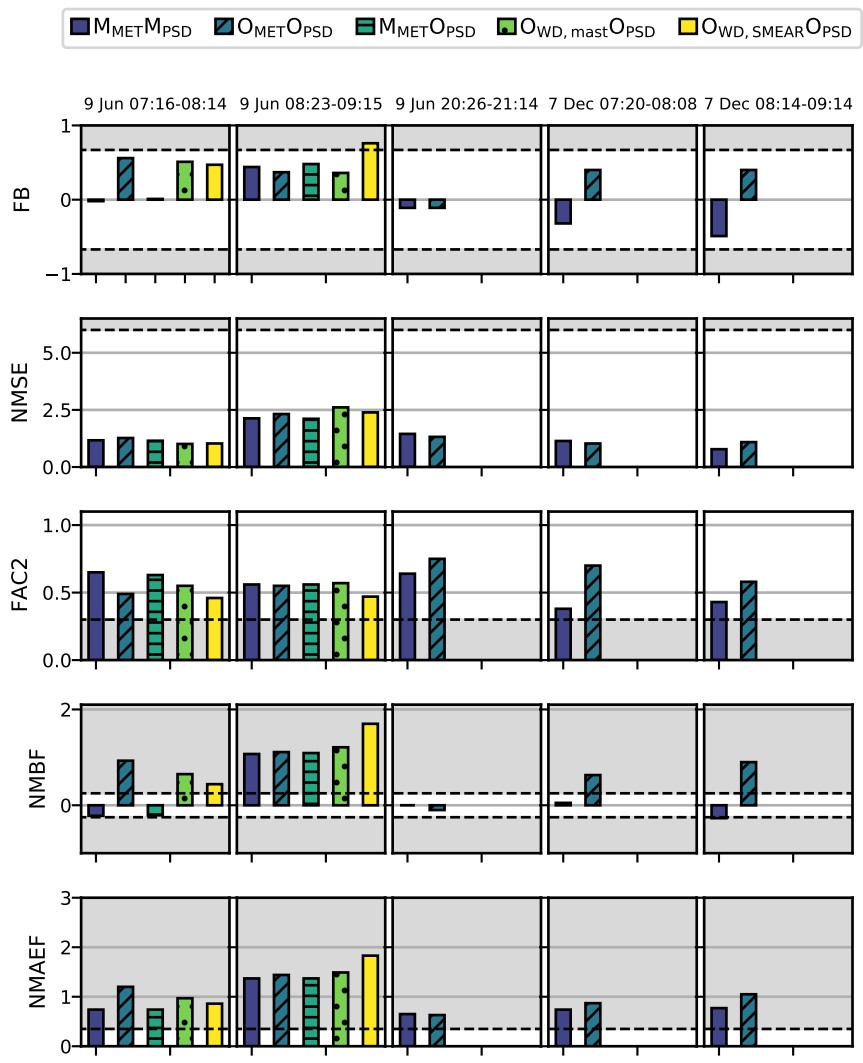

**Figure 4.** Model performance for the horizontal distribution of $N_{\text{tot}}$ using the performance measures fractional bias (FB), normalised mean squared error (NMSE), factor of two (FAC2), normalised mean bias factor (NMBF) and normalised mean absolute error factor (NMAEF). The grey indicates that the value exceeds the acceptance criteria given in Table 5. See Table 4 for the simulation names. Note that in the summer evening and winter morning, only two simulations have been conducted.

### 4.2.3 Aerosol size distribution

Fig. 9 illustrates the observed and modelled PSD for $\text{M}_{\text{MET}}\text{M}_{\text{PSD}}$ separately along the main and side street, at the supersite and opposite it, and in the background during the first hour of the summer morning 9 Jun. In addition to the Sniffer measurements with EEPS and ELPI, the modelled values are compared against DMPS measurements at the supersite and SMEAR III. The

5    model successfully reproduces PSD along the main street and specifically at the supersite, for which FB, NMSE and FAC2 are



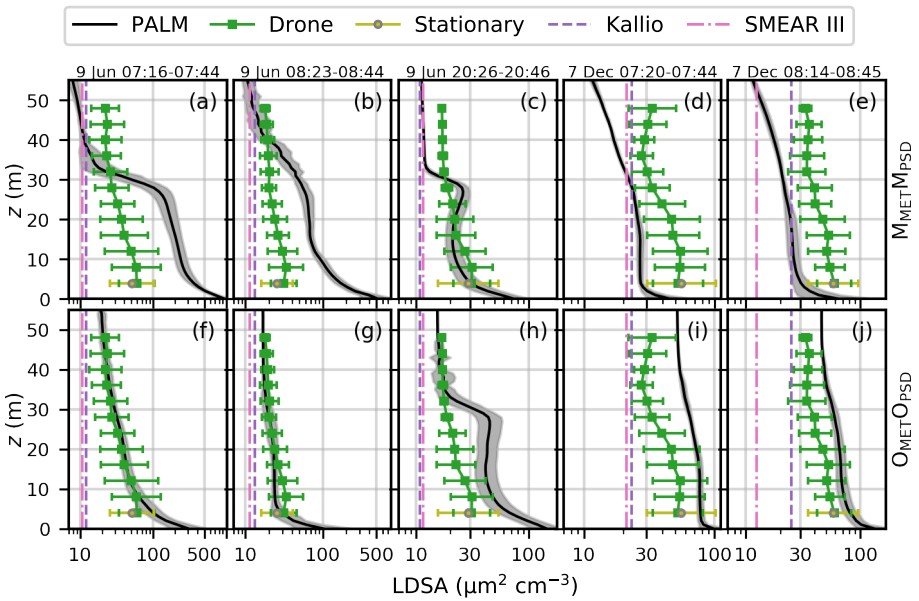

**Figure 5.** Measured (marker with error bar) and modelled (black line and grey shaded area) lung-deposited surface area (LDSA) of aerosol particles at the supersite for the summer morning (a,f for the first and b,g for the second hour), summer evening (c,h) and winter morning (d,i for the first and e,j for the second hour). The first row shows $M_{MET}M_{PSD}$ and the second $O_{MET}O_{PSD}$. The figure shows the geometric mean (measured: marker, modelled: solid line) and geometric standard deviation (measured: error bar, modelled: shaded area). Dashed lines show the geometric mean at the urban background monitoring sites in Kallio and SMEAR III.

within the acceptance criteria (see Table S3 in the Supplement). Also at the background, the Sniffer measurements agree with the model based on FB, NMSE and FAC2 even though the concentration of the smallest (the mean bin diameter $D_{mid} < 25\,\mathrm{nm}$) aerosol particles is underestimated. Instead along the side street, the modelled values are clearly lower than the observed and, for instance, based on NMBF the model underestimates the EEPS and ELPI observations by a factor of 3.45 and 5.48, respectively.

5    Comparing the two simulations with different boundary conditions, $M_{MET}M_{PSD}$ performs better along the main street and hence also at the supersite and opposite it during the first hour of the summer morning (Tables S3 and S4). However, during the second hour, the difference between $M_{MET}M_{PSD}$ and $O_{MET}O_{PSD}$ is minor, which was also observed for the horizontal distribution of $N_{tot}$. $O_{MET}O_{PSD}$, which uses the observed background PSD as boundary conditions, performs better along the side street and at the background. This is also observed in the summer evening (Tables S8 and S9) and winter morning

10    (Tables S10 and S11). In the summer evening, both $M_{MET}M_{PSD}$ and $O_{MET}O_{PSD}$ perform equally well along the main street and equally bad at the supersite overestimating the EEPS measurements by a factor of 3.68–4.36 based on NMBF (Tables S8 and S9). In the winter morning, $M_{MET}M_{PSD}$ overperforms along the main street, while at the supersite both $M_{MET}M_{PSD}$ and $O_{MET}O_{PSD}$ perform equally well and fulfil the acceptance criteria for FB, NMSE and FAC2 (Tables S10 and S11).





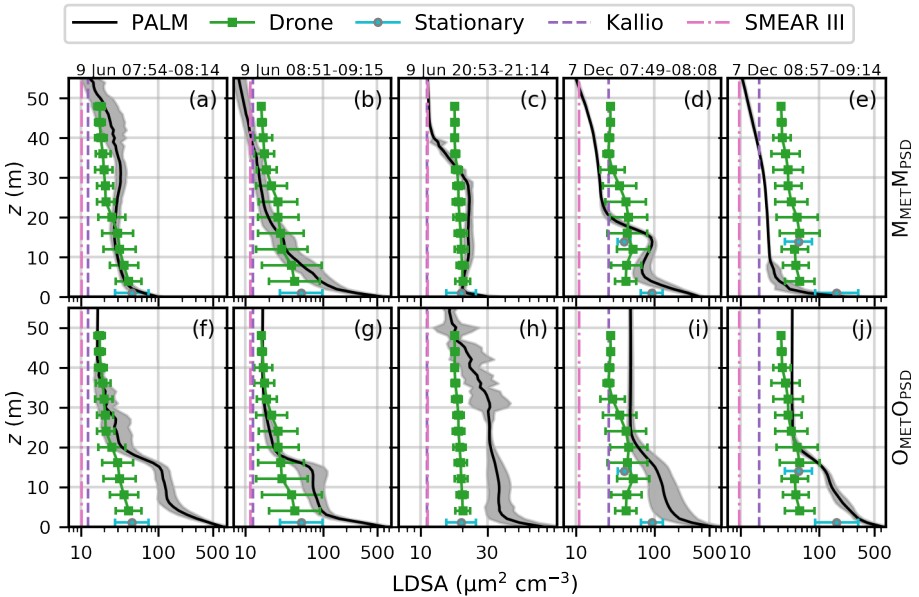

**Figure 6.** Measured (marker with error bar) and modelled (black line and grey shaded area) lung-deposited surface area (LDSA) of aerosol particles opposite the supersite for the summer morning (a,f for the first and b,g for the second hour), summer evening (c,h) and winter morning (d,i for the first and e,j for the second hour). The first row shows $M_{MET}M_{PSD}$ and the second $O_{MET}O_{PSD}$. The figure shows the geometric mean (measured: marker, modelled: solid line) and geometric standard deviation (measured: error bar, modelled: shaded area). Dashed lines show the geometric mean at the urban background monitoring sites in Kallio and SMEAR III.

### 4.2.4 Aerosol chemical composition

The chemical composition of aerosols was measured at the supersite by an ACSM and the black carbon (BC) concentration by a MAAP (see the Supplement, Table S1). Additionally Sniffer measured BC within $PM_1$. In general, the modelled and observed horizontal distribution of BC compare tolerably well based on the performance measures FB, NMSE and FAC2, while NMBF and NMAEF are not within the acceptance criteria (see Fig. S11 in the Supplement). Overall, the performance is best for $M_{MET}M_{PSD}$ in the summer morning during the first hour. In the summer and winter morning, $O_{MET}O_{PSD}$ suffers from high positive bias and absolute error and $M_{MET}M_{PSD}$ in the winter morning from high absolute error. Instead in the summer evening, both simulations show NMSE and FAC2 within the acceptance criteria but still overestimate BC. Comparing to the measured chemical composition of aerosols at the supersite (Tables 6 and S11-12) shows that the modelled concentration of organic carbon (OC) is in general in the right order of magnitude. Furthermore, in the summer morning especially the concentrations of sulphates ($SO_4^-$) and nitrates ($NO_3^{-1}$), and also BC and $PM_{2.5}$ are correctly reproduced by $O_{MET}O_{PSD}$, while ammonia is highly overestimated especially by $M_{MET}M_{PSD}$ (NMBF = 18.05). In the summer evening, $M_{MET}M_{PSD}$ overperforms $O_{MET}O_{PSD}$ and correctly reproduces $SO_4^-$, OC and $PM_{2.5}$, while overestimating the rest. Also in the winter morning, $M_{MET}M_{PSD}$ performs slightly better than $O_{MET}O_{PSD}$ modelling OC and $PM_{2.5}$ in the right order of magnitude but the other



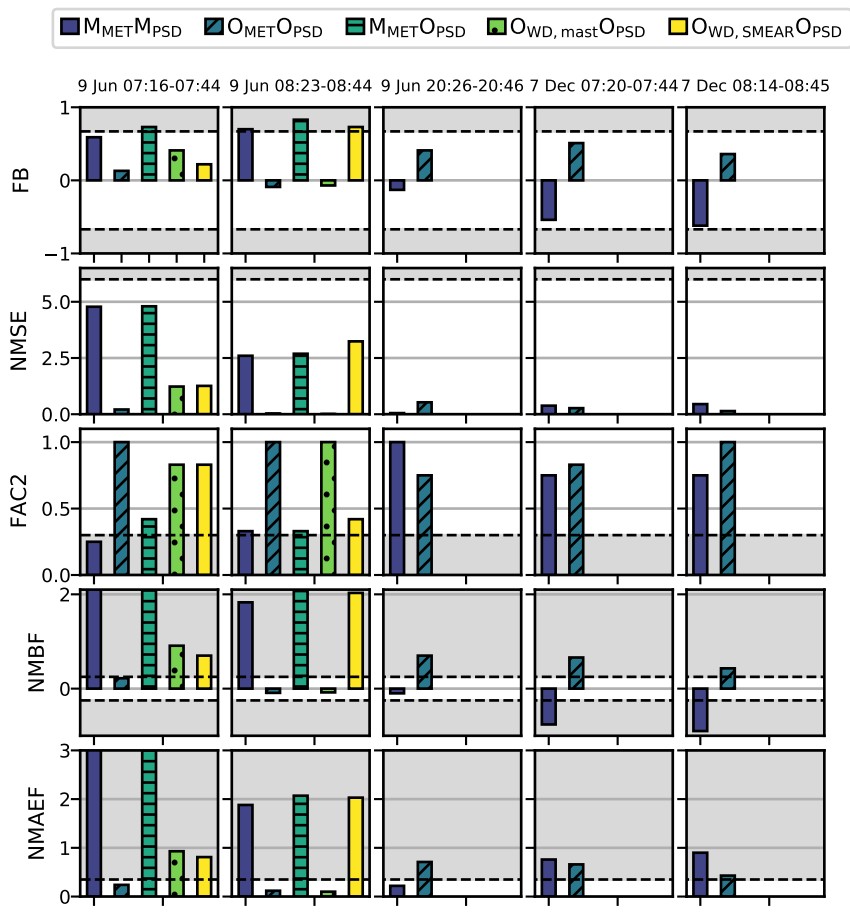

**Figure 7.** Model performance for the vertical distribution of LDSA at the supersite. The grey indicates that the value exceeds the acceptance criteria given in Table 5. See Fig. 4 for details.

chemical components are overestimated by a factor of around 2.7–4.2. Whether $M_{MET}M_{PSD}$ or $O_{MET}O_{PSD}$ performs better corresponds to the results on the vertical dispersion of LDSA.

### 4.3 Sensitivity

#### 4.3.1 Background aerosol size distribution

5    Regarding all variables used in the evaluation, only minor differences due to using modelled or measured PSD as a boundary condition are observed between $M_{MET}M_{PSD}$ and $M_{MET}O_{PSD}$ (e.g., FB = −0.02 and FB = 0.01, and NMSE = 1.17 and NMSE = 1.15 for the horizontal distribution of $N_{tot}$, respectively). The difference in the horizontal distribution of $N_{tot}$ (Fig. 10b) is mainly within -20–20 % with slightly higher (lower) concentrations in the southern (northern) part of the domain. This difference stems from roughly 80 % lower observed than modelled background $N_{tot}$ and thus the air being advected





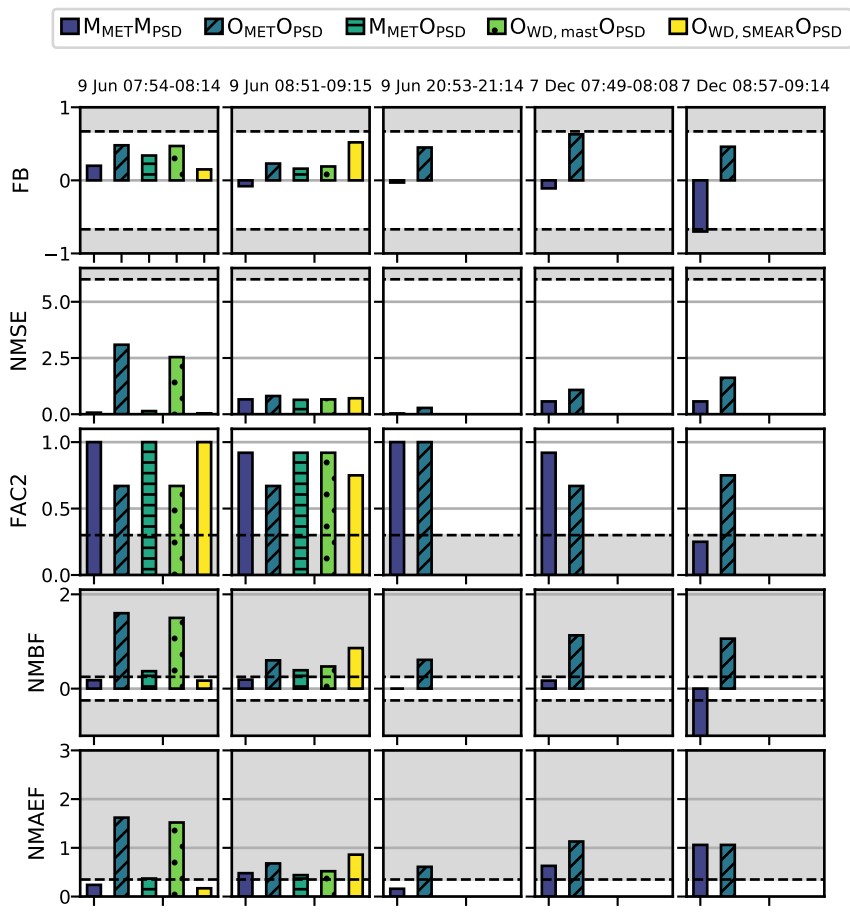

**Figure 8.** Model performance for the vertical distribution of LDSA opposite the supersite. The grey indicates that the value exceeds the acceptance criteria given in Table 5. See Fig. 4 for details.

from northwest is cleaner in $M_{MET}O_{PSD}$ compared to $M_{MET}M_{PSD}$. Similar to the horizontal distribution, the vertical profile of LDSA for $M_{MET}O_{PSD}$ does not differ from $M_{MET}M_{PSD}$ within the street canyon (Fig. 11). Only a small decrease in model performance is observed opposite the supersite when applying the observed PSD as the boundary condition (e.g., FB is increased from 0.20 to 0.34 and NMSE from 0.07 to 0.14 during the first hour, Fig. 8). However, above $z > 30$ m, the difference gradually approaches 65–160 %, i.e., the relative difference in the background $N_{tot}$ between the modelled ADCHEM values and SMEAR III observations. With respect to PSD, $M_{MET}M_{PSD}$ and $M_{MET}O_{PSD}$ perform mostly equally good or bad, except for slightly better performance of $M_{MET}O_{PSD}$ on the side street. The difference in the modelled PSD between $M_{MET}M_{PSD}$ and $M_{MET}O_{PSD}$ is smaller than when the wind speed and/or direction are different ($O_{WD,mast}O_{PSD}$ and $O_{WD,SMEAR}O_{PSD}$, Fig. S12 in the Supplement).



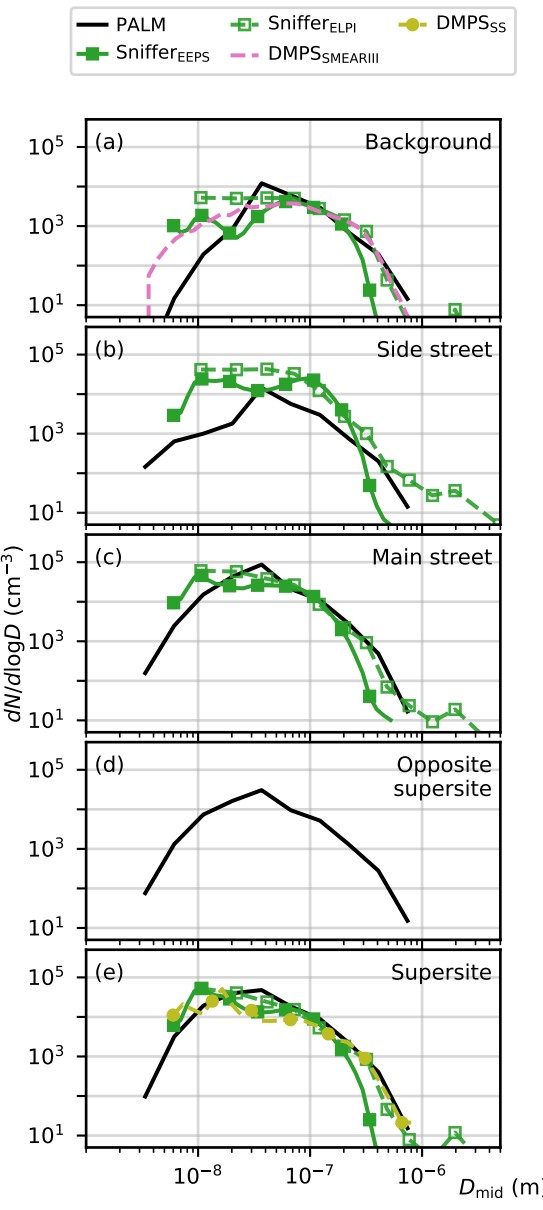

**Figure 9.** The mean aerosol number size distribution $dN/d\log D$ $(\mathrm{cm}^{-3})$ at different parts of the domain at $z = 1.5$ m for $\mathrm{M_{MET}M_{PSD}}$ on 9 Jun morning at 07:16–08:14. Modelled values are shown with a black solid line and Sniffer measurements with green lines: solid with filled squares for EEPS and dotted with empty squares for ELPI. Stationary DMPS measurements are shown with a light-green solid line with circles (SS = supersite) and pink dotted line (SMEAR III). Note that for this observation period no stationary Sniffer measurements are available opposite the supersite.





**Table 6.** Performance in the modelling aerosol chemical composition at the supersite on 9 Jun morning between 07:16–09:15 am. See Fig. 4 for further description.

| Simulation name | Variable | FB | NMSE | FAC2 | NMBF | NMAEF |
|---|---|---|---|---|---|---|
| $M_{MET}M_{PSD}$ | $SO_4^-$ | 0.73 | 1.65 | 0.50 | 1.60 | 1.60 |
| | OC | 0.42 | 0.75 | 0.50 | 0.75 | 0.97 |
| | $NO_3^-$ | 0.15 | 0.04 | 1.00 | 0.16 | 0.16 |
| | $NH_4^+$ | 1.60 | 26.44 | 0.00 | 18.05 | 18.05 |
| | BC | 1.39 | 15.07 | 0.00 | 10.02 | 10.02 |
| | $PM_{2.5}$ | 1.20 | 6.65 | 0.33 | 5.10 | 5.10 |
| $O_{MET}O_{PSD}$ | $SO_4^-$ | 0.42 | 0.26 | 1.00 | 0.54 | 0.58 |
| | OC | 0.13 | 0.06 | 1.00 | 0.14 | 0.21 |
| | $NO_3^-$ | 0.27 | 0.12 | 1.00 | 0.33 | 0.33 |
| | $NH_4^+$ | 1.06 | 4.02 | 0.00 | 3.02 | 3.02 |
| | BC | 0.50 | 1.69 | 0.50 | 1.09 | 1.28 |
| | $PM_{2.5}$ | 0.69 | 0.86 | 0.50 | 1.15 | 1.15 |

### 4.3.2 Background meteorological conditions

In Section 4.2, the simulation using the observed data as boundary conditions ($O_{MET}O_{PSD}$) was shown to perform worse than $M_{MET}M_{PSD}$ in the summer morning. As the observed wind speed at Kivenlahti and the one modelled by MEPS differ (see Section 4.1), we separately investigate the influence of the incoming wind direction on the model sensitivity.

In general, $O_{MET}O_{PSD}$ and $O_{WD,mast}O_{PSD}$, for which the incoming wind direction is replaced with the one measured on the Kivenlahti mast but the wind speed is the same, result in a similar pattern for the difference in the horizontal distribution of $N_{tot}$ compared to $M_{MET}M_{PSD}$ (Fig. 10a,c). However, the differences are larger for $O_{MET}O_{PSD}$, for which the incoming upper-level wind speed is up to $2\,\mathrm{m\,s^{-1}}$ slower during the first hour (Fig. 2a). This results in the aerosol particles being transported more to the southwest side of the main street. As the wind is more from the north in $O_{MET}O_{PSD}$ than in $O_{WD,mast}O_{PSD}$,

also the impact of wind direction on the street canyon vortex along the main street is observed by clearly lower (higher) $N_{tot}$ on the southern (northern) side of the street canyon. Instead $O_{WD,SMEAR}O_{PSD}$, for which the wind direction is from SMEAR III and does not vary with height, shows clearly higher concentrations (up to +150 %) along the main street but smaller differences in its surroundings (-40–80 %). Replacing the modelled wind direction with the one observed on the Kivenlahti mast ($O_{WD,mast}O_{PSD}$) or SMEAR III ($O_{WD,SMEAR}O_{PSD}$) improves the model performance for the horizontal distribution of $N_{tot}$

during the first summer morning hour (Fig. 4). However, for the second hour $O_{WD,SMEAR}O_{PSD}$ is shown to perform even worse than $O_{MET}O_{PSD}$ based on a lower FAC2 and higher NMBF and NMAEF. During the first hour, $O_{WD,SMEAR}O_{PSD}$ performs better than $O_{WD,mast}O_{PSD}$ based on FB, NMBF and NMAEF, indicating that there is more bias in $O_{WD,mast}O_{PSD}$, while during the second hour $O_{WD,SMEAR}O_{PSD}$ performs better only based on NMSE.



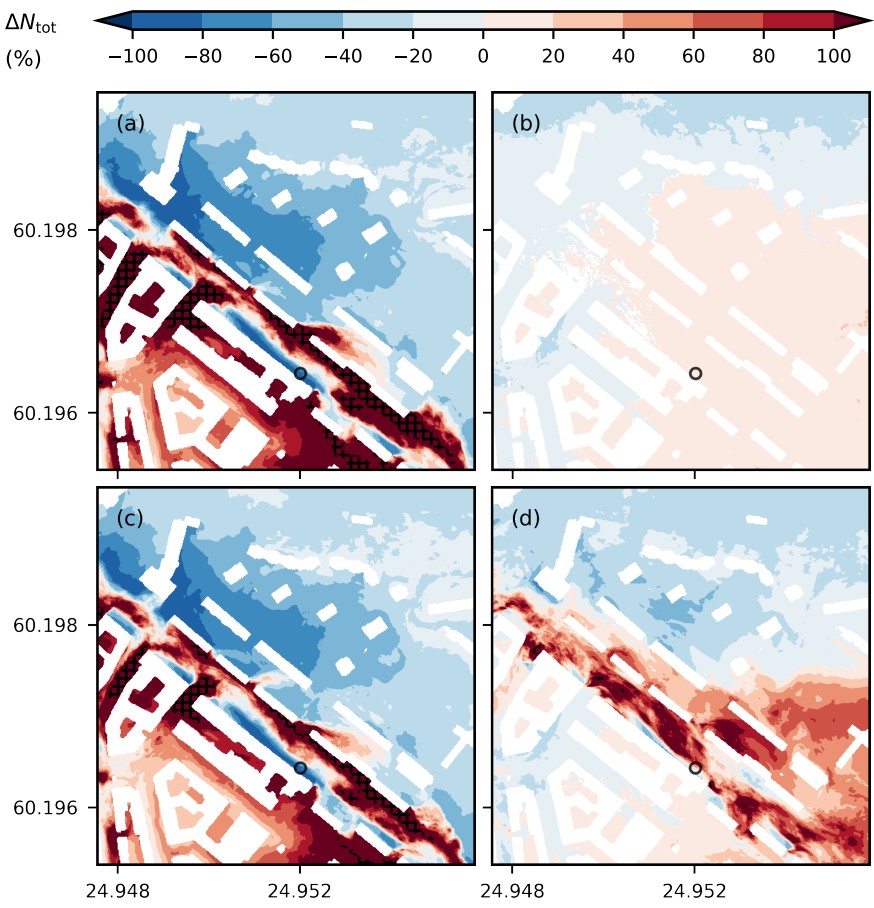

**Figure 10.** Relative difference in the total aerosol number concentration ($\Delta N_{\text{tot}}$) at $z = 2.5$ m on 9 Jun morning between 07:16–09:15 am compared to $\text{M}_{\text{MET}}\text{M}_{\text{PSD}}$ for a) $\text{O}_{\text{MET}}\text{O}_{\text{PSD}}$, b) $\text{M}_{\text{MET}}\text{O}_{\text{PSD}}$, c) $\text{O}_{\text{WD,mast}}\text{O}_{\text{PSD}}$ and d) $\text{O}_{\text{WD,SMEAR}}\text{O}_{\text{PSD}}$. Area with black crosses show $|\Delta N_{\text{tot}}| >$ 150 %. Buildings are shown with white and black circle denotes the location of the supersite. Note that $\Delta N_{\text{tot}}$ is shown here for the whole simulation and not separately for each hour.

At the supersite (Figs. 11a,c and 7), $\text{O}_{\text{MET}}\text{O}_{\text{PSD}}$ agrees better than $\text{M}_{\text{MET}}\text{M}_{\text{PSD}}$ with the observations (e.g., FB $= 0.13$ and FB $= 0.59$, and NMSE $= 0.21$ and NMSE $= 4.78$, respectively, during the first hour) and hence modifying the MEPS wind direction to correspond the observed one at Kivenlahti increases the performance (Fig. 7). During the second hour, $\text{O}_{\text{WD,mast}}\text{O}_{\text{PSD}}$ performs even slightly better than $\text{O}_{\text{MET}}\text{O}_{\text{PSD}}$ (e.g., FB $= -0.07$ and FB $= -0.09$, and NMSE $= 0.01$ and NMSE $= 0.03$).

5    However, applying the wind direction from SMEAR III improves the model performance only for the first hour while during the second hour $\text{O}_{\text{WD,SMEAR}}\text{O}_{\text{PSD}}$ performs worst. Where $\text{O}_{\text{WD,mast}}\text{O}_{\text{PSD}}$ results in lower (higher) LDSA below (above) $z = 30 - 40$ m, during the second hour $\text{O}_{\text{WD,SMEAR}}\text{O}_{\text{PSD}}$ shows higher values until the building height ($z = 19$ m), after which it gradually starts to follow the $\Delta$LDSA for $\text{O}_{\text{WD,mast}}\text{O}_{\text{PSD}}$. Opposite the supersite up to five-fold values compared to $\text{M}_{\text{MET}}\text{M}_{\text{PSD}}$ are observed in $\text{O}_{\text{MET}}\text{O}_{\text{PSD}}$ and $\text{O}_{\text{WD,mast}}\text{O}_{\text{PSD}}$ within the first hour (Fig. 11b) and the model performance is clearly decreased





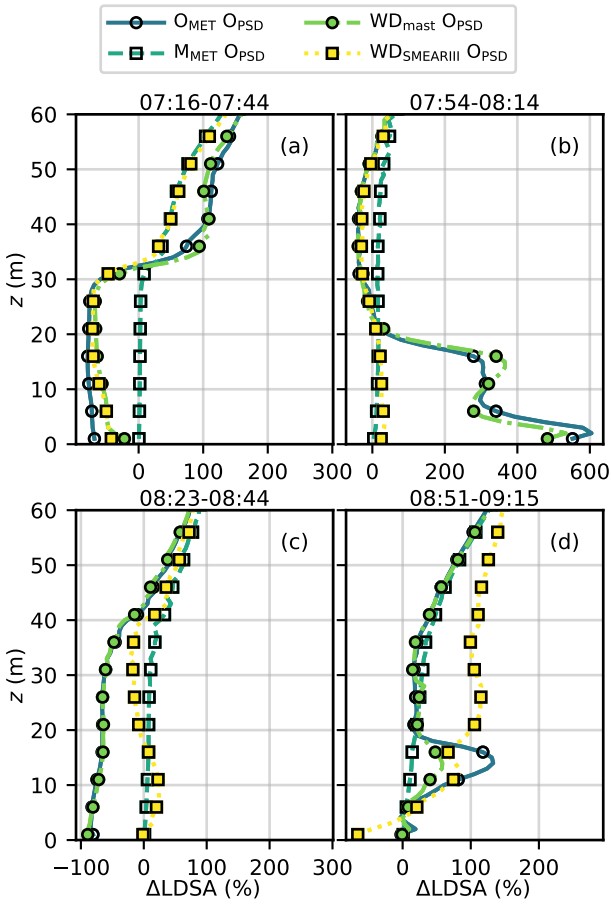

**Figure 11.** Relative difference in the lung-deposited surface area ($\Delta$LDSA) of aerosol particles compared to $M_{MET}M_{PSD}$ at the supersite (a,c) and opposite supersite (b,d) on 9 Jun morning. The figure shows the difference in the geometric mean for $O_{MET}O_{PSD}$ (solid line with empty circles), $M_{MET}O_{PSD}$ (dashed line with empty squares), $O_{WD,mast}O_{PSD}$ (dash-dot line with filled squares) and $O_{WD,SMEAR}O_{PSD}$ (dotted line with filled squares).

when using the observed wind direction from Kivenlahti (e.g., FB is increased from 0.20 to 0.47 and FAC2 decreased from 1.00 to 0.67, Fig. 8). Instead, $O_{WD,SMEAR}O_{PSD}$ performs better than $M_{MET}M_{PSD}$. Within the second hour, $\Delta$LDSA is mainly within $\pm 50$ % below the building height, but above $O_{WD,SMEAR}O_{PSD}$ deviates from the other profiles showing two-fold values. All $\Delta$LDSA profiles gradually approach $\sim 100$ %, which results from using different PSD for $M_{MET}M_{PSD}$ and for rest of the

5   simulations.

Similarly for the aerosol size distribution, changing the modelled wind direction by MEPS to the one measured at Kivenlahti improves model performance at the background and slightly decreases elsewhere during the first hour, whereas during the last hour PSD is modelled better along the side street and at the supersite (see Tables S3 and S6 in the Supplement). Applying the wind direction from SMEAR III generally does not improve model performance (Table S7). Along the main street, the





difference in PSD is mainly governed by the emission, which is shown by the peak at $D_{\mathrm{mid}} \approx 30$ nm in Fig. S12 (in the Supplement), while at the background $\Delta N$ follows the difference in the boundary conditions for aerosol particles.

## 5   Discussion and conclusions

This study provides an extensive evaluation of the SALSA2.0 module in the PALM model system 6.0 on simulating the
horizontal distribution of aerosol particle number concentrations ($N_{\mathrm{tot}}$), size distributions and black carbon concentrations and the vertical distribution of surface area (LDSA) in a complex urban environment. In addition, the aerosol chemical composition in a single measurement location is examined. Simulations are conducted under three different meteorological conditions: two hours in a summer morning, one hour in a summer evening and two hours in a winter morning. The study also investigates the model sensitivity to the boundary conditions of meteorological variables and background aerosol concentrations during the
different times.

     In the summer morning, a stronger temperature inversion is present in the MEPS model data compared to the Kivenlahti observations. Still, $N_{\mathrm{tot}}$ along the main street is higher when measured boundary conditions ($\mathrm{O_{MET}O_{PSD}}$) are used instead of the modelled ones ($\mathrm{M_{MET}M_{PSD}}$) during the first hour. This likely stems from the underestimation of the wind speeds above 217 m in $\mathrm{O_{MET}O_{PSD}}$, leading to lower mechanical turbulence production and mixing. During the second hour, no large differences in
$N_{\mathrm{tot}}$ are observed, as the wind speed and direction of the input data become more equal. The 45–90° difference in the modelled and measured wind directions especially during the first hour results in pollutant accumulation and overestimation of LDSA at the supersite (i.e., to the southwestern side of the main street canyon) in $\mathrm{M_{MET}M_{PSD}}$ and opposite the supersite in $\mathrm{O_{MET}O_{PSD}}$. This also leads to overestimation of especially nitrates and black carbon at the supersite in $\mathrm{M_{MET}M_{PSD}}$, whereas the chemical composition of aerosol particles is well reproduced in $\mathrm{O_{MET}O_{PSD}}$. However in terms of PSD, $\mathrm{M_{MET}M_{PSD}}$ in general performs
better than $\mathrm{O_{MET}O_{PSD}}$ around the main street and at the supersite.

     By the evening, MEPS shows only slightly more southerly but clearly stronger winds at $z < 200$ m than what is observed on the Kivenlahti mast. To be precise, MEPS predicts a low-level jet with the maximum wind speed at $z = 100$ m. Still, both simulations perform nearly equally. Slightly higher $N_{\mathrm{tot}}$ and overestimation of near-surface LDSA in $\mathrm{O_{MET}O_{PSD}}$ can be explained by the difference in the wind speed. This is also reflected in the aerosol chemical composition at the supersite.
MEPS predicts a more stable stratification, which might justify why the difference in the spatial variability of aerosol particle concentrations between $\mathrm{M_{MET}M_{PSD}}$ and $\mathrm{O_{MET}O_{PSD}}$ is not that huge.

     MEPS predicts clearly stronger wind speeds in the winter morning, but still the near-surface $N_{\mathrm{tot}}$ is mainly overestimated in $\mathrm{O_{MET}O_{PSD}}$ during the first simulation hour. The higher concentrations in $\mathrm{O_{MET}O_{PSD}}$ can be attributed to a stronger temperature inversion leading to less efficient mixing. During the second hour, the difference in performance is small. Similarly, opposite
the supersite $\mathrm{O_{MET}O_{PSD}}$ clearly overestimates LDSA below the building height, but interestingly $\mathrm{O_{MET}O_{PSD}}$ performs better in modelling the vertical distribution of LDSA at the supersite. $\mathrm{M_{MET}M_{PSD}}$ slightly underestimates the concentrations at the supersite, but also the background concentration of LDSA is lower. Interestingly, the difference in the chemical composition is



not systematic as $M_{MET}M_{PSD}$ reproduces better the concentration of organics and $O_{MET}O_{PSD}$ the concentration of nitrates and black carbon.

Modifying the boundary conditions from MEPS by applying the wind direction measured on the Kivenlahti mast ($O_{WD,mast}O_{PSD}$) improves model performance in the summer morning, indicating that the wind predicted by MEPS is not correct. Especially the

vertical profiles of LDSA are sensitive to the wind direction as it influences the formation and direction of the canyon vortex and thus the accumulation of pollutant to the leeward side of the street canyon. The similar patterns of $\Delta N_{tot}$ for $O_{MET}O_{PSD}$ and $O_{WD,mast}O_{PSD}$ but the higher absolute values of $\Delta N_{tot}$ for $O_{MET}O_{PSD}$ indicate that the horizontal distribution is strongly controlled by the wind direction, while the absolute values depend on the wind speed. In contrast, applying the vertically constant wind direction from SMEAR III worsens the model performance within the second simulation hour. Presumably wind has

too much westerly component compared to the northerly winds in the MEPS and Kivenlahti data, which results in the traffic-emissions downstream being flushed along the main street and also being accumulated near the ground. Contrary to the wind direction, the background PSD has only a small influence on the model performance. However, the shape of PSD influences also the magnitude of integrated aerosol measures, such as LDSA, and therefore, for example, the background LDSA is over three-fold in $M_{MET}M_{PSD}$ compared to $O_{WD,mast}O_{PSD}$ in the summer evening.

Consequently, meteorological boundary conditions are particularly important for quantitative urban air quality modelling using LES, and therefore the inlet meteorology should be evaluated prior to conducting CFD simulations (Santiago et al., 2020). However in our case, we are unable to evaluate the modelled meteorology. In Santiago et al. (2020), the meteorological observations were made within the simulation domain, while in our case the closest measurements at SMEAR III are 800 m away from the supersite. Observations are available also from the Kivenlahti mast, which has several measurement levels, but is

located over 17 km away from the supersite and represents more semi-urban to rural area. Another problem with the Kivenlahti data is the lack of wind observations above 217 m in summer, which presumably leads to, for instance, underestimation of the incoming wind speed during the first simulation hour in the summer morning and around 9 pm in the summer evening. Consequently, neither observations are optimal for evaluating the modelled meteorology nor providing meteorological boundary conditions for the simulations.

Of the aerosol metrics applied, LDSA directly estimates the health effect of aerosol exposure. The mean modelled LDSA concentration at $z = 4$ m varies between 27–360 $\mu m^2\,cm^{-3}$ at the supersite and 20–250 $\mu m^2\,cm^{-3}$ opposite the supersite, with the overall lowest LDSA opposite the supersite in $M_{MET}M_{PSD}$ in the summer evening and highest at the supersite in $M_{MET}M_{PSD}$ in the summer morning. As mentioned above, the wind direction is shown determining for accumulation of pollutants near the ground. The difference in near-ground LDSA between the supersite and opposite the supersite is the most

pronounced in $M_{MET}M_{PSD}$ during the first hour of the summer morning (360 and 37 $\mu m^2\,cm^{-3}$, respectively). This large concentration gradient across the street illustrates the degree of error that can be made in the estimated outdoor-exposure level in epidemiological studies. Compared to urban background and traffic monitoring stations (see Kuula et al., 2020, and references within), a street canyon allows for strong accumulation leading to high instantaneous concentrations.

However, LDSA is often overestimated near the ground in our simulations. One limitation of this study and in general

in urban LES is omitting vehicle-induced turbulence (VIT), which would enhance vertical pollutant transport and mixing





near the surface and very likely decrease concentrations near ground. The research to include VIT in LES without extensive computational costs is on-going and currently no freely available VIT-model exists for LES. Neglecting the thermal turbulence in the simulations is another important limitation of our study. We acknowledge that omitting the influence of anthropogenic heat and heating by incoming solar radiation leads to overestimation of the vertical stability near the ground, which can

partly explain the overestimation of the modelled surface concentrations. However, the spatial variability has been shown less dependent on a detailed heating distribution Nazarian et al. (2018) and therefore the horizontal distribution is mainly determined by the predominant inflow conditions. Lastly, condensation of the biogenic volatile organic compounds on aerosol particles and their consecutive growth are not considered in PALM.

*Code and data availability.* The PALM code is freely available under the GNU General Public License v3. The exact model source code

(revision 4416) is available at https://palm.muk.uni-hannover.de/trac/browser?rev=4416 (last access: 10 Sept 2019). MEPS model data is distributed under Norwegian license for public data (NLOD) and Creative Commons 4.0 BY Internasjonal at http://thredds.met.no/thredds/catalog.html by the Norwegian Meteorological Institute.

All measurement data applied in the evaluation can be downloaded from https://doi.org/10.5281/zenodo.3828508 (Kurppa et al., 2020) and the input and output data, performance measures and source code modifications from https://doi.org/10.5281/zenodo.3824351 (Kurppa,

2020a). The scripts applied in the data analysis and model evaluation are freely available at https://doi.org/10.5281/zenodo.3839462 (Kurppa, 2020b), and the files and scripts for creating the PALM input data at https://doi.org/10.5281/zenodo.3839684 (Kurppa and Strömberg, 2020).

*Video supplement.* TEXT

## Appendix A: Performance measures

Performance measures calculated using the modelled $M_i$ and observed $O_i$ values. $N$ = number of samples.

$$\mathrm{FB} = \frac{1}{N} \sum \frac{(M_i - O_i)}{0.5(M_i + O_i)} \tag{A1}$$

$$\mathrm{NMSE} = \frac{N \sum (M_i - O_i)^2}{\sum M_i \sum O_i} \tag{A2}$$

FAC2 = fraction of data that satisfy

$$0.5 \leq \frac{M_i}{O_i} \leq 2.0 \tag{A3}$$





$$\text{NMBF} = \frac{\sum(M_i - O_i)}{\sum O_i} \quad , \text{if } \overline{M} \geq \overline{O} \tag{A4}$$

$$= \frac{\sum(M_i - O_i)}{\sum M_i} \quad , \text{if } \overline{M} < \overline{O} \tag{A5}$$

$$\text{NMAEF} = \frac{\sum|M_i - O_i|}{\sum O_i} \quad , \text{if } \overline{M} \geq \overline{O} \tag{A6}$$

$$= \frac{\sum|M_i - O_i|}{\sum M_i} \quad , \text{if } \overline{M} \geq \overline{O} \tag{A7}$$

*Author contributions.* LJ and MK designed the concept of the study. MK prepared and conducted the LES simulations with contributions from AH, and PR conducted the ADCHEM simulations. MK, JS and SK contributed to the pre-processing of the used input data sets. LJ, HK, TR, JN, LP and HT planned the measurement campaign and MK, HK, SK and HT conducted parts of the measurements. AB participated to the post-processing scripts of the Sniffer data. MK wrote the manuscript with contributions from all co-authors.

*Competing interests.* The authors declare that they have no conflict of interest.

*Acknowledgements.* The authors are very grateful to Aleksi Malinen and Sami Kulovuori from the Metropolia University of Applied Sciences for technical expertise and operation of Sniffer, and to Aeromon Oy and Helsinki Region Environmental Services Authority (HSY) for collaboration in conducting the drone measurements. This study was financially supported by the Doctoral programme in Atmospheric Sciences (ATM-DP), Helsinki Metropolitan Region Urban Research Program, the Academy of Finland Centre of Excellence (no. 307331), Business Finland (Cityzer; Services for effective decision making and environmental resilience) and Project Smart urban solutions for air

quality, disasters and city growth (SMURBS, no. 689443) funded by ERA-NET-COFUND project under ERA-PLANET.



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
