# Peer review of "Sensitivity of spatial aerosol particle distributions to the boundary conditions in the PALM model system 6.0"

_Geoscientific Model Development, 2020_

## Referee Comment (RC1) · Anonymous Referee #1 · 7 Aug 2020

Review of gmd-2020-163

Sensitivity of spatial aerosol particle distributions to the boundary conditions in the PALM model system 6.0

The study applies the LES model PALM in an urban setup in order to model air quality parameters in a street canyon of a major road and its surrounding in Helsinki. The model is evaluated against stationary and mobile observations. Furthermore, the authors examine different ways of providing boundary conditions for their LES simulations. The evaluation of the different simulations is conducted by applying a number of different statistical measures. This detailed evaluation is scientifically sound. However,

the complexity of several different statistical measures in addition to the different simulations makes it difficult to write the analysis in an understandable way, which would make it easy for the reader to follow the analysis and results. I highly recommend to revise sections 4.2 and 4.3 in terms of writing. In these sections, the authors often jump between one measure or simulation and the other. For some of the paragraphs and sentences it was not clear to which simulation or time period they were referring to. Overall, I don't have concerns about the scientific relevance and the quality of the applied analysis. Therefore, the manuscript can be accepted for publication after minor revision.

Specific comments p.3, l.8-9: What do the authors mean with "but not necessarily stable performance"? If you apply boundary conditions from other model runs, you often use a (mostly coarser) larger scale model run. Such a continuous run would usually enable continuous boundary data. The statement also seems to be in contrast to p.6, l.28-29 ("which allows realistic [...] boundary conditions").

p.6., l.29: Do the authors mean that the forcing mesoscale flow does not provide enough turbulence, which would take time and distance to be generated on the higher resolution LES domain? I suggest to explain this with one or two more sentences to the benefit of the readers.

p.8, l.2-4: Are only the mast observations used as observation-based driving data? I think, the description of how observational data serve as boundary conditions could be extended.

p.8, l.31-32: Can the authors provide the size distribution parameters of the Hietikko et al. (2018) size distribution in the manuscript?

p.9, l.9: There is no number given for EFPM in the manuscript. Can it be included in the manuscript? Furthermore, do EFPM and EFPM2.5 refer to the same quantity? In case not, I think it would help the reader to follow the calculation if numbers for both are given.

p.10, l.14-15: Would it be possible to run including variable roughness length, e.g. derived from the building structure or density? If so, would you expect a strong impact on the air flow? If it can be expected that a variable roughness length may affect the results substantially, I suggest to include this consideration in the discussion section.

p.11, l.14-15: In the figures the observations above 200m show lower temperature and higher dew point temperature than the model.

p.11, l.28-30: For 21:00-24:00, there are two ADCHEM peaks, whereas the smaller one is at ~50nm(?) and the second at 100nm, which is matching the observations.

Figures S5, S10: Related to comment above. If this isn't a huge effort, I think it would be helpful to summarize the modelled PNSDs matching the investigation periods, i.e. 7:00-10:00 and 20:00-22:00, instead of the currently given time windows. Same is for winter period shown in Fig. S10.

p.12, l.3-4, Figure S10: The observed peak seems to be clearly at smaller sizes than the modelled one during 9:00-12:00. Again, providing the PNSD comparison for 7:00-10:00 would be helpful. Perhaps the authors could put these in the paper, and leave the diurnal evolution as it is in the supplement?

p.13, l.1-2: Can the authors please explain in little more detail the meaning of these acceptance criteria and why the have chosen these thresholds? That is, what error / deviation is accepted if the criteria is fulfilled. This is done exemplary later in the text for NMBF and NMAEF, however, I think it would be helpful already here in a more general manner.

p.16, l.4: In Table S3, I find NMBF for the side street in the summer morning first hour of -2.45 and -4.58 for EEPS and ELPI, respectively, which is different from the numbers reported in the text. Can the authors check again if the numbers in the text are correct and if so, please explain how these were calculated based on NMBF?

p.16, l.11: Related to the comment above, the numbers 3.68-4.36 differ from NMBF in

the tables S8 and S9.

p.18, l.1: Overestimated by MMETMPSD or OMETOPSD? Are the numbers 2.7 and 4.2 supposed to be found in Table 6 or not shown?

p.18, l.1-2: What do the authors mean with the last sentence of this section? I think this thought needs more explanation in the text.

p.21., l.8: Slower than what? It is not clear to what other observation or model result the wind speed of OMETOPSD is compared to.

p. 24, l.24-26: Can the authors please explain this thought and its conclusion?

Other comments p.1, l.10: I suggest to change "factor of two" to "fraction of data within a factor of two", or similar.

p.1, l.12-13: One "and" too much in the enumeration.

p.2, l.7: What do the authors mean with "being at the same level"? I assume the authors refer to the height, i.e. in ~1.5m above ground? However, the pollutants are not only in this height, as the sentence might suggest.

p.2, l.34 – p.3, l.1: Does this sentence refer to the study by Kuurpa et al. 2019? If so, the link to the sentence before is not clear.

p.3, l.16-17: I think, better English would be "can model boundary conditions cause" -> "can be / is caused by model boundary conditions".

p.3., l.21-23: Perhaps try to split this rather long sentence holding so much information into two or more sentences to improve readability.

p.7, Table 2: I did not do the maths, but just see that two numbers deviate. Below the table it says "591 m", however Lz is given as 606 m in the table. Which is the correct one?

p.8, l.28 & 29: Should it be "total mass emission factors" and "number emission factors",

at least this is what EF usually stands for. Emission factor is also used in the following.

p.8, l.30: "sensitivity" -> "sensitive".

p.12, Figure 2; Figure S6; Figure S8: Missing ")" after "m s-1".

p.14, Figure 3: Is the tick mark in the yellow color at 2 x 105 cm-3?

Figure 4, 7 and 8: I suggest to change in the caption "The grey ..." to "The grey area ...", or similar.

p.16, l.2: Is "overperform" the right word? To my non-native speaker knowledge it means something like "better than expected"? In a quick search, I only find it in a financial context.

p.17, l.3 & 4: The abbreviations ACSM and MAAP should be given with their long name here.

p.17, l.9: The statement refers to Figures S12 and S13, not "S11-12".

p.17, l.11: Nitrate has a typo superscript "1".

p.17, l.12: NH4+, as named in Table 6 is ammonium not ammonia. Furthermore, similar to the other substances the chemical formula for ammonium should be given in brackets in the text.

p.17, l.12-13: Again, I don't understand the meaning of "overperform" her. Do the authors mean something like "performs better than OMETOPSD"?

p.17, Figure 6: Really unimportant detail... Nevertheless, the color for "Stationary" changed compared to Figure 5? I think it is easier for the reader if it would not.

p.18, l.7: I suggest to add "see Fig. 4" somewhere in this brackets.

p.21, Table 6: Can the authors color code the numbers as done for Table S12 and S13?

p.21, Table 6; Table S12, Table S13: I suggest to change "Performance in the modelling" to "Performance of the modelled" or "Model performance for" in the table captions.

p.22, l.1: It should be mentioned in the beginning of this paragraph that you now focus on the performance for LDSA.

p.22, l.2: I suggest to add "of the summer morning" after "during the first hour".

p.22, l.6: I think instead of "Where" it should be "Whereas".

p.22, l.6: Does "lower" refer to "lower than MMETMPSD"? If so, it should be mentioned here.

p.23, Figure 11: It should be OWD,mastOPSD and OWD,SMEAROPSD in the legend.

p.23, l.2-3: It should be mentioned that it is now refered to the difference to MMETMPSD and not the observations.

Supplement p.1: The seems to be something wrong with the section counting. S1 is missing, also S3.2 and S3.3.

---

## Short Comment (SC1) · 27 Aug 2020

This is an executive editor comment highlighting the ways in which this manuscript is not currently compliant with GMD policy on code and data availability. In this case, most of the code and data availability is excellent. There are just a couple of issues:

1. The PALM server is insufficiently persistent in exactly the same way that GitHub is insufficiently persistent. PALM is GPL so you are simply allowed to archive the version you used somewhere better. The easiest thing to do would be to grab a tarball of the correct revision and upload it to Zenodo.

[Figure]

2. Thredds is rather unfortunate because it's hard to cite well, however in this case the citation is just a link to the top level of the server. This wouldn't be enough for someone to find the data you used. Please find a way to be more precise. At a trivial level, the link needs to be https rather than http.

Further details on code and data availability requirements are in the GMD model code and data policy: https://www.geoscientific-model-development.net/about/code_and_data_policy.html. The reasons for the policy and more detail are provided in this editorial: https://doi.org/10.5194/gmd-12-2215-2019.
* * *

---

## Author Comment (AC1) · 28 Aug 2020

Dear Executive Editor,

Thank you for your comments. The exact model version (PALM model system 6.0, revision 4416) has been uploaded to Zenodo and can be downloaded from https://doi.org/10.5281/zenodo.4005366. The "Code and data availability" section of the manuscript will be modified accordingly.

---

## Referee Comment (RC2) · Anonymous Referee #2 · 7 Sep 2020

The manuscript (MS) deals with evaluating a new model system that couples an aerosol dynamics module (SALSA2.0) with a LES model system (PALM 6.0) for high-resolution urban air quality modeling. The main objective is to validate the horizontal and vertical distributions of aerosols in terms of particle number concentrations, size distributions and chemical compositions. In particular, authors investigate the model sensitivity to meteorological boundary conditions and aerosol background concentrations.

The authors simulated three periods in summer and winter with different meteorological conditions and compared them with the measurements conducted using a mobile

laboratory and a drone in an urban neighborhood in Helsinki, Finland. The results highlight the high sensitivity of urban LES modeling to meteorological boundary conditions and the aerosol background concentrations.

The methods and assumptions are scientifically sound and well explained in the MS. The outcomes have important implications for future studies urban air quality modeling using LES. I consider the objectives of the MS interesting for the community and within the scopes of the journal. However, the presentation need improvements. Therefore, I recommend the MS for publication after minor revision. My major points are:

1- In the result section, there are extensive and detailed explanations (sometimes too wordy) about the plots but no discussion. There are few sentences in section 5 but it is not enough. The authors should move the discussions to section 4 and expand them. It is important to explain "what" we see in the plots. Nevertheless, more important than that is to know "why".

2- The model somehow struggles with the mixing state of the atmosphere. I want to see direct quantitative measure of turbulence (e.g., TKE) at least between different runs. The vertical profile of the potential temperature or the Richardson number could be helpful too. Most importantly, the discrepancies are attributed to the mixing state in the simulations and observations. Thus, a direct measure of the atmospheric mixing state would be essential.

3- Several statistics are used in the MS but it is not clear what they represent. In the current form, they are rather confusing and make it hard to grasp the key message. For the horizontal distribution, I recommend SAL method.

4- I am a bit confused with the role of the aerosol dynamics and chemistry. It seems that the simulations differ in boundary conditions only. But the PSD and composition differ too. So is there a feedback from the atmospheric state to aerosol dynamics and chemistry? With the chemical boundary conditions fixed, these differences stem from the processes within the child domain only. Is that right? What are the individual roles

of aerosol dynamics, chemistry and meteorology? It would be helpful to elaborate on this.

Other comments:

P1L8: do you mean "are driven" instead of "are drawn"? This occurs several times in the MS.

P4L25: please explain in detail why you choose these dates. I assume it is based on the diurnal and seasonal variations in the mixing state of the atmosphere. What about the urban heat island?

P6L19: wouldn't this be part of the reason why OC is well captured but not sulfate and Nitrate? What about the winter period?

P6L23: "at the same time"?

P6L23: "a high enough resolution is needed" is too generic. Please add a range. Table 2: the innermost domain has 1 m resolution. But later the results are aggregate a 5 m grid for comparisons. Then what is the point of this expensive simulation?

P9L22-26: this text is repetition of table 4.

P10L20: The boundaries of the innermost domain are fixed for the chemicals. This means that the air masses come and go without bringing or taking any pollutants. Does this make sense in the resolution you are dealing with?

P11L5: Most of the figures cited here are in the supplementary material. This is not helpful for the reader. I understand that the MS should not be lengthy. But perhaps with some reorganization

Figure 2: Adding the potential temperature to this plot would be helpful.

Figure 3: add name tags to each row and column so that the reader can navigate more easily.

Figure 4: I have problem understanding this type of figure. Perhaps because the purpose of each parameter is not well explained? What about SAL method?

Figure 5: It is difficult to have a solid conclusion here. Higher LDSA is (or should be) associated with higher number concentration. The model always fails to capture the profile in the morning hours. What are the individual contributions of MET and PSD?

P17L1-10: same as the previous comment.

P19L7-8: this is an odd sentence.

Figure 9: It might be that the coagulation (aerosol dynamics) is not fast enough. Is this a reason why fine particles are overestimated? This can be tested by aerosol dynamics on/off.

Figure 10 and Table S2: SAL might be a better method to compare these plots.

P24L26: replace "huge" with "large".

―――――――――――――――

---

## Author Comment (AC2) · 18 Sep 2020

Dear reviewer,

Please find as a supplement our point-by-point response to the Referee comments and also a file showing the changes made in the manuscript.

Please also note the supplement to this comment:
https://gmd.copernicus.org/preprints/gmd-2020-163/gmd-2020-163-AC2-supplement.zip

---

## Author Response (AR1)

**Referee response to "Sensitivity of spatial aerosol particle distributions to the boundary conditions in the PALM model system 6.0" by Mona Kurppa et al.**

We thank both referees for their valuable comments and suggestions. Please find our detailed point-by-point responses below (in black).

The changes made to the manuscript are visualised in the attached file "manuscript_see_differences.pdf". Page and line numbers given in this response refer to that document.

P = page

L = line number

Anonymous Referee #1:

The study applies the LES model PALM in an urban setup in order to model air quality parameters in a street canyon of a major road and its surrounding in Helsinki. The model is evaluated against stationary and mobile observations. Furthermore, the authors examine different ways of providing boundary conditions for their LES simulations. The evaluation of the different simulations is conducted by applying a number of different statistical measures. This detailed evaluation is scientifically sound. However, the complexity of several different statistical measures in addition to the different simulations makes it difficult to write the analysis in an understandable way, which would make it easy for the reader to follow the analysis and results. I highly recommend to revise sections 4.2 and 4.3 in terms of writing. In these sections, the authors often jump between one measure or simulation and the other. For some of the paragraphs and sentences it was not clear to which simulation or time period they were referring to. Overall, I don't have concerns about the scientific relevance and the quality of the applied analysis. Therefore, the manuscript can be accepted for publication after minor revision.

Thank you for this comment regarding the clarity and readability. The Results section has now separated into three separate sections: 4. Comparison of the modelled and observed boundary conditions, 5. Evaluation of the air quality modelling results and 6. Sensitivity analysis. Furthermore, discussion about the meaning of the results has been moved from the section Discussion and conclusions to the respective results sections (see the comments of the Reviewer #2 and especially the major point 1). The changes have been marked in blue in the manuscript with track changes.

Specific comments

p.3, l.8-9: What do the authors mean with "but not necessarily stable performance"? If you apply boundary conditions from other model runs, you often use a (mostly coarser) larger scale model run. Such a continuous run would usually enable continuous boundary data.

The statement also seems to be in contrast to p.6, l.28-29 ("which allows realistic [...] boundary conditions").

With this phrase we wanted to point out that NWP models do not perform equally well in all weather conditions. To make this idea more clear, the phrase has been modified as follows:

*"... which provide a good spatial coverage but not necessarily stable performance."* --> "... which provide a good spatial coverage but not necessarily stable performance **in all prevailing weather conditions**." (P3 L9)

p.6., l.29: Do the authors mean that the forcing mesoscale flow does not provide enough turbulence, which would take time and distance to be generated on the higher resolution LES domain? I suggest to explain this with one or two more sentences to the benefit of the readers.

Yes, this is correct. The following sentences were added:

"*To reduce the time and distance for the mesoscale flow field to adjust to the LES modelling domain, a synthetic turbulence generator within PALM can be applied.*" --> "**As the mesoscale data do not contain resolved-scale turbulence, turbulence must first be developed within the PALM domain.** To reduce the time and distance for the mesoscale flow field to adjust **and turbulence to develop** within the LES modelling domain, a synthetic turbulence generator within PALM can be applied." (P6 L30-32)

p.8, l.2-4: Are only the mast observations used as observation-based driving data? I think, the description of how observational data serve as boundary conditions could be extended.

The following sentences were added:

"Meteorological observations from the SMEAR~III station at $z = 31$ m are downloaded using the SmartSMEAR tool (Junninen et al., 2009)." (P8 L1-2)

"When applying the SMEAR III data, constant values are used for the entire vertical profile." (P8 L8-9)

p.8, l.31-32: Can the authors provide the size distribution parameters of the Hietikko et al. (2018) size distribution in the manuscript?

A figure on the aerosol size distribution has now been added to the Supplement (P7, Fig. S4). It includes the aerosol size distribution measured by Hietikko et al. (2018) and the one applied for the traffic-combustion-related aerosol emission in this study.

p.9, l.9: There is no number given for $EF_{PM}$ in the manuscript. Can it be included in the manuscript? Furthermore, do $EF_{PM}$ and $EF_{PM2.5}$ refer to the same quantity? In case not, I think it would help the reader to follow the calculation if numbers for both are given.

$EF_{PM}$ has now been included in Table 3. Furthermore, on $EF_{PM2.5}$ has been replaced by $EF_{PM}$ on P9 L1.

p.10, l.14-15: Would it be possible to run including variable roughness length, e.g. derived from the building structure or density? If so, would you expect a strong impact on the air flow? If it can be expected that a variable roughness length may affect the results substantially, I suggest to include this consideration in the discussion section.

In PALM, the Monin-Obukhov similarity theory (MOST) is applied as the wall model between the surface and the first grid level where scalars and horizontal velocity components are defined. This requires providing the roughness length, which characterises the roughness elements not resolved by the computational grid. As these simulations apply a high grid resolution (1-9 m), the roughness length describes mainly the surface material, while the impact of building structure and packing density on the flow are explicitly resolved.

p.11, l.14-15: In the figures the observations above 200 m show lower temperature and higher dew point temperature than the model.

Thank you for pointing out this typo. The sentences have been modified as follows:

"*The observed and modelled profiles of air (T) and dew-point temperature ($T_D$) correspond qualitatively well (Figs. 2c and S4 in the Supplement), but the observations higher values of T and $T_D$ than MEPS above z = 200 m, especially at 8-9 am. Hence, MEPS predicts a stronger and shallower temperature inversion, which would lead to weaker vertical mixing.*"
--> "The observed and modelled profiles of air (T) and dew-point temperature ($T_D$) correspond qualitatively well (Figs. 2c and S4 in the Supplement), but the observations show **lower (higher)** values of **T ($T_D$)** than MEPS above z = 200 m, especially at 8-9 am. MEPS **also** predicts a stronger and shallower **surface** temperature inversion, which would lead to weaker vertical mixing." (P11 L16-19)

p.11, l.28-30: For 21:00-24:00, there are two ADCHEM peaks, whereas the smaller one is at~50 nm(?) and the second at 100 nm, which is matching the observations.

The previous Figure S5 showed the size distributions in UTC time. Hence, the one matching the summer evening simulation would be 18:00-21:00.

We have now added the measured and modelled (ADCHEM) aerosol number size distributions for the specific modelling times to the Supplement (Fig. S6, S9 and S12). Fig. S9 for the evening simulation shows that the modelled PSD peaks at around 87 nm and the measured at 70 nm. These have now been corrected to the manuscript as well (P12 L1).

Figures S5, S10: Related to comment above. If this isn't a huge effort, I think it would be helpful to summarize the modelled PNSDs matching the investigation periods, i.e. 7:00-10:00 and 20:00-22:00, instead of the currently given time windows. Same is for winter period shown in Fig. S10.

Thank you for the comment. Background PSDs are now provided only for the specific modelling periods (see Fig. S6, S9 and S12).

p.12, l.3-4, Figure S10: The observed peak seems to be clearly at smaller sizes than the modelled one during 9:00-12:00. Again, providing the PNSD comparison for 7:00-10:00 would be helpful. Perhaps the authors could put these in the paper, and leave the diurnal evolution as it is in the supplement?

See the comments above. Note that the previous background PSD figures were provided in UTC time, not the local time which is UTC+3.

p.13, l.1-2: Can the authors please explain in little more detail the meaning of these acceptance criteria and why the have chosen these thresholds? That is, what error /deviation is accepted if the criteria is fulfilled. This is done exemplary later in the text for NMBF and NMAEF, however, I think it would be helpful already here in a more general manner.

The acceptance criteria are based on Hanna and Chang (2012) and Yu et al. (2010). For more details, the reader is suggested to look for these publications (see Table 5 caption).

p.16, l.4: In Table S3, I find NMBF for the side street in the summer morning first hour of -2.45 and -4.58 for EEPS and ELPI, respectively, which is different from the numbers reported in the text. Can the authors check again if the numbers in the text are correct and if so, please explain how these were calculated based on NMBF?

The values in the text are correct. From Yu et al. 2006, p. 29:

*"For example, $B_{NMBF}$ can be interpreted as follows: if $B_{NMBF}$ is positive, the model overestimates the observations by a factor of $B_{NMBF}+1$; e.g. for $B_{NMBF} = 1.2$, the model overestimates the observations by a factor of 2.2. If $B_{NMBF}$ is negative, the model underestimates the observations by a factor of $1-B_{NMBF}$; for example, $B_{NMBF} = -1.2$ indicates that the model underestimates the observations by a factor of 2.2."*

p.16, l.11: Related to the comment above, the numbers 3.68-4.36 differ from NMBF in the tables S8 and S9.

See the response above.

p.18, l.1: Overestimated by $M_{MET}M_{PSD}$ or $O_{MET}O_{PSD}$? Are the numbers 2.7 and 4.2 supposed to be found in Table 6 or not shown?

By $M_{MET}M_{PSD}$. This has now been clarified in the phrase:

*"Also in the winter morning (Table S13), $M_{MET}M_{PSD}$ performs slightly better than $O_{MET}O_{PSD}$ modelling OC and $PM_{2.5}$ in the right order of magnitude, but the other chemical components are overestimated by a factor of around 2.7–4.2."* --> "Also in the winter morning (Table S13), $M_{MET}M_{PSD}$ performs slightly better than $O_{MET}O_{PSD}$ in modelling OC and $PM_{2.5}$ in the right order of magnitude, but $M_{MET}M_{PSD}$ still overestimates the mass concentrations of the other chemical components by a factor of around 2.7–6.5." (P22 L4-7)

The phrase refers to Table S13. To improve readability, references to the correct tables have now been added (P22 L5).

p.18, l.1-2: What do the authors mean with the last sentence of this section? I think this thought needs more explanation in the text.

Thank you for the comment. The sentence has been improved as follows:

*"Whether $M_{MET}M_{PSD}$ or $O_{MET}O_{PSD}$ performs better corresponds to the results on the vertical dispersion of LDSA."* --> Comparing modelled values with point observations in a street canyon is very sensitive to the correct wind direction because perpendicular wind component leads to accumulation of pollutants to the leeward side of the street canyon. As the vertical dispersion of LDSA was also shown sensitive to the wind direction, the results on the performance of modelling the correct chemical composition corresponds to those on the vertical dispersion of LDSA (see Section 4.2.2)." (P22 L7-11)

p.21, l.8: Slower than what? It is not clear to what other observation or model result the wind speed of $O_{MET}O_{PSD}$ is compared to.

Thank you for pointing out this. "than in $O_{WD,mast}O_{PSD}$ was added." (P23 L25-26)

p. 24, l.24-26: Can the authors please explain this thought and its conclusion?

Stronger wind speeds generate more turbulence, which enhances ventilation and dispersion of pollutants upwards from the street level. Stronger atmospheric stability, instead, suppresses turbulence and ventilation of pollutants, leading to higher street-level concentrations.

In the summer evening simulation, MEPS data shows higher wind speeds (--> more turbulence and enhanced ventilation), but also stronger stratification (--> weaker turbulence and weaker ventilation) than the observation at the Kivenlahti mast. Hence, the impact of stronger wind speed and stronger stratification can balance each other, which could explain similar results in $M_{MET}M_{PSD}$ and $O_{MET}O_{PSD}$.

Other comments

p.1, l.10: I suggest to change "factor of two" to "fraction of data within a factor of two", or similar.

"fraction of data within a " was added for clarity (P1 L10).

p.1, l.12-13: One "and" too much in the enumeration.

There is a comma missing that separates the two sentences: The horizontal distribution is most sensitive to the wind speed and atmospheric stratification, and vertical distribution (is sensitive) to the wind direction. A comma was added (P1 L13).

p.2, l.7: What do the authors mean with "being at the same level"? I assume the authors refer to the height, i.e. in~1.5 m above ground? However, the pollutants are not only in this height, as the sentence might suggest.

We changed "level" to "height" (P2 L7). As it is stated in the phrase, traffic exhaust and road dust are emitted around the same height (i.e., 1 m above ground) where urban dwellers inhale outdoor air.

p.2, l.34 – p.3, l.1: Does this sentence refer to the study by Kuurpa et al. 2019? If so,the link to the sentence before is not clear.

A citation to Kurppa et al. (2019) was added for clarity (P2 L34)

p.3, l.16-17: I think, better English would be "can model boundary conditions cause" ->"can be / is caused by model boundary conditions".

Thank you for the comment. The phrase was modified as follows:

*"Hence, it is still unclear how much uncertainties in aerosol particle concentrations and size distributions can model boundary conditions cause."* --> "Hence, it is still unclear how much uncertainty in aerosol particle concentrations and size distributions is caused by model boundary conditions." (P3 L16-17)

p.3., l.21-23: Perhaps try to split this rather long sentence holding so much information into two or more sentences to improve readability.

Thank you for the comment. This long phrase was modified as follows:

*"The campaign focused on the spatial variability of aerosol particle number, surface area and mass both in horizontal and vertical as well as aerosol size distributions and chemical composition with a high temporal and spatial resolution measured using a mobile laboratory and a drone."* ---> "The campaign focused on the spatial variability of aerosol particle number, surface area and mass both in horizontal and vertical as well as aerosol size distributions and chemical composition. The observations were carried out with a high temporal and spatial resolution using a mobile laboratory and a drone." (P3 L21-23)

p.7, Table 2: I did not do the maths, but just see that two numbers deviate. Below the table it says "591 m", however Lz is given as 606 m in the table. Which is the correct one?

Thank you for noticing this typo. The value 606 m is the correct one and the table footnote has been corrected (P7).

p.8, l.28 & 29: Should it be "total mass emission factors" and "number emission factors", at least this is what EF usually stands for. Emission factor is also used in the following.

You are correct. The word "factor" has been added accordingly:

"Aerosol particle emission inventories are typically provided as total mass emission **factors** $EF_{PM2.5}$. In SALSA, these would need to be translated to number emission **factors** $EF_N$, assuming some size distribution for the emitted aerosol particles." (P9 L1-2)

p.8, l.30: "sensitivity" -> "sensitive".

Thanks! This has now been corrected. (P9 L3)

p.12, Figure 2; Figure S6; Figure S8: Missing ")" after "m s$^{-1}$".

Thank you for noticing these! Missing brackets have been added accordingly.

p.14, Figure 3: Is the tick mark in the yellow color at $2 \times 10^5$ cm$^{-3}$?

Yes, this is correct. A tick label at $2 \times 10^5$ cm$^{-3}$ has been added.

Figure 4, 7 and 8: I suggest to change in the caption "The grey ..." to "The grey area...", or similar.

"The grey indicates" has been modified to "The grey **area** indicates" in these figure captions.

p.16, l.12: Is "overperform" the right word? To my non-native speaker knowledge it means something like "better than expected"? In a quick search, I only find it in a financial context.

For clarity, the word "overperform" is no longer used in the manuscript. The phrases containing "overperform" have been modified as follows:

*"In the winter morning, $M_{MET}M_{PSD}$ fulfills the acceptance criteria during the first hour, except for NMAEF, and overperforms $O_{MET}O_{PSD},...$"* --> "In the winter morning, $M_{MET}M_{PSD}$ fulfills the acceptance criteria during the first hour, except for NMAEF, and performs better than $O_{MET}O_{PSD}...$" (P16 L6-7).

*"In the winter morning, $M_{MET}M_{PSD}$ overperforms along the main street..."* --> "In the winter morning, $M_{MET}M_{PSD}$ produces better results than $O_{MET}O_{PSD}$ along the main street." (P19 L 8)

*"In the summer evening, $M_{MET}M_{PSD}$ overperforms $O_{MET}O_{PSD}$ and correctly reproduces..."* --> "In the summer evening, $M_{MET}M_{PSD}$ corresponds better to observations than $O_{MET}O_{PSD}$ and correctly reproduces..." (P22 L3-4)

p.17, l.3 & 4: The abbreviations ACSM and MAAP should be given with their long name here.

Both ACSM and MAAP have now been written out in the text (see P20 L1-2).

p.17, l.9: The statement refers to Figures S12 and S13, not "S11-12".

Thank you for noticing this typo. It has been corrected accordingly (P20 L10).

p.17, l.11: Nitrate has a typo superscript "1".

This typo has now been corrected.

p.17, l.12: NH4+, as named in Table 6 is ammonium not ammonia. Furthermore, similar to the other substances the chemical formula for ammonium should be given in brackets in the text.

Thank you for noticing this typo. It has been corrected on P22 L2. Furthermore, $NH_3$ was also falsely named as ammonium instead of ammonia on P6 L13.

p.17, l.12-13: Again, I don't understand the meaning of "overperform" here. Do the authors mean something like "performs better than $O_{MET}O_{PSD}$"?

See the comment above. The word "overperform" is no longer used in the manuscript for clarity.

p.17, Figure 6: Really unimportant detail... Nevertheless, the color for "Stationary" changed compared to Figure 5? I think it is easier for the reader if it would not.

It is important to make the manuscript as easy as possible to follow. The same colour has now been used for "Stationary" in Figs. 6 and 7.

p.18, l.7: I suggest to add "see Fig. 4" somewhere in this brackets.

The phrase has now been modified as follows (P22 L16 – P23 L1-4).

*"Regarding all variables used in the evaluation, only minor differences due to using modelled or measured PSD as a boundary condition are observed between $M_{MET}M_{PSD}$ and $M_{MET}O_{PSD}$ (e.g., FB = −0.02 and FB = 0.01, and NMSE = 1.17 and NMSE = 1.15 for the horizontal distribution of N tot , respectively)."* --> "Regarding all variables used in the evaluation, only minor differences due to using modelled or measured PSD as a boundary condition are observed between $M_{MET}M_{PSD}$ and $M_{MET}O_{PSD}$.  For instance for the horizontal distribution of $N_{tot}$ (e.g., FB = −0.02 and FB = 0.01, and NMSE = 1.17 and NMSE = 1.15 for the horizontal distribution of $N_{tot}$, respectively (see Fig. 4). "

p.21, Table 6: Can the authors color code the numbers as done for Table S12 and S13?

We must check this with the Editiorial Support whether it is possible to use colour-coded fonts in the main text.

p.21, Table 6; Table S12, Table S13: I suggest to change "Performance in the modelling" to "Performance of the modelled" or "Model performance for" in the table captions.

*"Performance in the modelling"* has been changed to "Performance of the modelled" in the captions of Table 6 (P22), and Tables S12 and S13 (P21 in the supplement).

p.22, l.1: It should be mentioned in the beginning of this paragraph that you now focus on the performance for LDSA.

The phrase was modified as follows:

*"At the supersite (Figs. 11a,c and 7), $O_{MET}O_{PSD}$ agrees better than $M_{MET}M_{PSD}$ with the observations (e.g., FB = 0.13 and FB = 0.59, and NMSE = 0.21 and NMSE = 4.78, respectively, during the first hour) and hence modifying the MEPS wind direction to correspond the observed one at Kivenlahti increases the performance (Fig. 7)."* --> "The observed vertical profile of LDSA at the supersite on the summer morning corresponds better to the modelled by $O_{MET}O_{PSD}$ and $O_{WD,mast}O_{PSD}$ than $M_{MET}M_{PSD}$ (Fig. 7). Hence,

modifying the MEPS wind direction to correspond the observed one at Kivenlahti increases the model performance." (P23 L9-14)

 I suggest to add "of the summer morning" after "during the first hour".

"on the summer morning" was added to the phrase on P24 L9 (see the comment above).

 I think instead of "Where" it should be "Whereas".

This is correct. This has been corrected accordingly (P25 L2).

 Does "lower" refer to "lower than $M_{MET}M_{PSD}$"? If so, it should be mentioned here.

This is corrected. "than $M_{MET}M_{PSD}$" has been added (P25 L2).

 It should be $O_{WD,mast}O_{PSD}$ and $O_{WD,SMEAR}O_{PSD}$ in the legend.

This has now been corrected (P25).

 It should be mentioned that it is now refered to the difference to $M_{MET}M_{PSD}$ and not the observations.

"compared to $M_{MET}M_{PSD}$" has been added to the phrase (P25 L8).

Supplement p.1: The seems to be something wrong with the section counting. S1 is missing, also S3.2 and S3.3.

The section counting follows the one of the main text. Hence, some sections which do not include any supplementary material, are missing from the counting.

Anonymous Referee #2:

The manuscript (MS) deals with evaluating a new model system that couples an aerosol dynamics module (SALSA2.0) with a LES model system (PALM 6.0) for high-resolution urban air quality modeling. The main objective is to validate the horizontal and vertical distributions of aerosols in terms of particle number concentrations, size distributions and chemical compositions. In particular, authors investigate the model sensitivity to meteorological boundary conditions and aerosol background concentrations.

The authors simulated three periods in summer and winter with different meteorological conditions and compared them with the measurements conducted using a mobile laboratory and a drone in an urban neighborhood in Helsinki, Finland. The results highlight the high sensitivity of urban LES modeling to meteorological boundary conditions and the aerosol background concentrations.

The methods and assumptions are scientifically sound and well explained in the MS. The outcomes have important implications for future studies urban air quality modeling using LES. I consider the objectives of the MS interesting for the community and within the scopes of the journal. However, the presentation need improvements. Therefore, I recommend the MS for publication after minor revision. My major points are:

1- In the result section, there are extensive and detailed explanations (sometimes too wordy) about the plots but no discussion. There are few sentences in section 5 but it is not enough. The authors should move the discussions to section 4 and expand them. It is important to explain "what" we see in the plots. Nevertheless, more important than that is to know "why".

Thank you for this valuable comment. The Results section has now separated into three separate sections: 4. Comparison of the modelled and observed boundary conditions, 5. Evaluation of the air quality modelling results and 6. Sensitivity analysis. Furthermore, discussion about the meaning of the results has been moved from the section Discussion and conclusions to the respective results sections. The changes have been marked in blue in the manuscript with track changes.

2- The model somehow struggles with the mixing state of the atmosphere. I want to see direct quantitative measure of turbulence (e.g., TKE) at least between different runs. The vertical profile of the potential temperature or the Richardson number could be helpful too. Most importantly, the discrepancies are attributed to the mixing state in the simulations and observations. Thus, a direct measure of the atmospheric mixing state would be essential.

The vertical profiles of TKE for the child domain have been now included for comparison (PXX Fig. X). Discussion about this figure has been added in the following locations:

"This likely stems from the underestimation of the wind speeds above 217 m in $O_{MET}O_{PSD}$ (Fig. 2a), leading to lower mechanical turbulence production and mixing (Fig. 5a on the turbulent kinetic energy (TKE)). " (P15 L1-3)

"This is surprising considering the clearly stronger winds in MEPS at z < 200 m than what is observed on the Kivenlahti mast. Yet, MEPS predicts a more stable stratification, which leads to nearly equal TKE values (Fig. 5b). This can justify why the difference in the spatial variability of aerosol particle concentrations between $M_{ME}OM_{PSD}$ and $O_{MET}O_{PSD}$ is not that large." (P16 L2-5)

"Contrary to the summer evening, MEPS predicts clearly lower wind speeds in the winter morning, which would lead to weaker mixing, but at the same time the observed temperature inversion on the Kivenlahti mast is stronger than the modelled by MEPS especially during the first hours. Hence, the stronger stability and suppression of turbulence (Fig. 5c) can explain the higher concentrations in $O_{MET}O_{PSD}$. " (P16 L8-11)

3- Several statistics are used in the MS but it is not clear what they represent. In the current form, they are rather confusing and make it hard to grasp the key message. For the horizontal distribution, I recommend SAL method.

The statistical measures applied in this study are mostly well-known and frequently applied, except for NMBF and NMAEF which have been explained in more detail. However, we agree that the Results section could be improved. We have now added more discussion and explanations to Section 5.1 and Table 5. The modified text has been marked in the manuscript (P13 L11-16).

Thank you also for proposing the usage of SAL. We calculated the statistics, but we think that the added value is not notable. SAL has been created for evaluation model performance in modelling precipitation areas. Hence, this method could be valuable when modelling the dispersion of individual pollutant puffs. More reasoning:

- Here, L (location) is always small because the concentration fields are bounded by the buildings.

- A (amplitude) is already measured by NMSE and NMAEF.

- Since the evaluation is done at resolution of 5 m using both spatially and temporally averaged concentration fields, it can be misleading to analyse S. For example, strong gradients in the modelling can be greatly smoothed by the averaging.

4- I am a bit confused with the role of the aerosol dynamics and chemistry. It seems that the simulations differ in boundary conditions only. But the PSD and composition differ too. So is there a feedback from the atmospheric state to aerosol dynamics and chemistry? With the chemical boundary conditions fixed, these differences stem from the processes within the child domain only. Is that right? What are the individual roles of aerosol dynamics, chemistry and meteorology? It would be helpful to elaborate on this.

Thank you for the comment.

In this study, we wanted to focus on the impact of the boundary conditions on the simulated concentrations, as this information is essential for the model users. Also, a detailed analysis on the gaseous species has been left out and will likely be provided in a future study. This was done to avoid making the manuscript too wide.

The individual roles of different aerosol dynamic processes have been investigated in Kurppa et al. (2019).

Yes, the differences in concentrations stem from the chemical and physical processes of air pollutants only within the innermost (child) domain.

Other comments:

P1L8: do you mean "are driven" instead of "are drawn"? This occurs several times in the MS.

We use "are drawn from" on purpose. For clarity, we have replaced it with "are taken from" (see P1 L8 and P7 L6).

P4L25: please explain in detail why you choose these dates. I assume it is based on the diurnal and seasonal variations in the mixing state of the atmosphere. What about the urban heat island?

Dates were chosen so that the weather conditions clearly differ, but also based on the observation data quality and coverage.

P6L19: wouldn't this be part of the reason why OC is well captured but not sulfate and Nitrate? What about the winter period?

The transport of organic vapours from the chemistry module to SALSA and back is still under development and hence the impact of e.g. VOC on aerosol particle growth is not yet considered in (P28 L 23-24). We have now added the following phrase to discuss this linkage between the chemistry module and SALSA:

"However, the transfer of different organic vapours from the chemistry module to SALSA is still under development." (P6 L20-21)

P6L23: "at the same time"?

Thank you for noticing this typo. It has now been corrected to "at the same time" (P6 L24).

P6L23: "a high enough resolution is needed" is too generic. Please add a range.

We added "in the order of ~1 m" for the range (P6 L24).

Table2: the innermost domain has 1 m resolution. But later the results are aggregate a 5 m grid for comparisons. Then what is the point of this expensive simulation?

The mobile observation data are aggregated to a resolution of 5 m for two reasons: 1) the GPS signal has an accuracy in the order of 5 m and 2) the number of observation points is limited and therefore in a 1 m x 1 m grid one grid point could represent only one measurement instance.

Applying this resolution (5 m) for the model is not fine enough for resolving the flow in street canyons (Xie and Castro, 2006).

P9L22-26: this text is repetition of table 4.

Thank you for the comment. We still think that it is the clearest to explain the simulation names in the text as well as list them in a table.

P10L20: The boundaries of the innermost domain are fixed for the chemicals. This means that the air masses come and go without bringing or taking any pollutants. Does this make sense in the resolution you are dealing with?

In the time scales considered here (1-2 hours), we can assume that the background concentrations remain rather the same. The aerosol and chemistry modules are run only within the innermost domain to limit the computational costs, as these modules make the simulations 10-20 times more expensive.

P11L5: Most of the figures cited here are in the supplementary material. This is not helpful for the reader. I understand that the MS should not be lengthy. But perhaps with some reorganization

As you mentioned, the manuscript becomes easily very lengthy. We have carefully decided the figures we think are the most relevant for transforming the message and supporting the conclusions.

Figure 2: Adding the potential temperature to this plot would be helpful.

We decided to keep only absolute temperature in the figure. However, TKE figure has been added (P16 Fig. 5).

Figure 3: add name tags to each row and column so that the reader can navigate more easily.

Name tags for each row (measured or modelled) and column (times) have been added (P14 Fig. 3).

Figure 4: I have problem understanding this type of figure. Perhaps because the purpose of each parameter is not well explained? What about SAL method?

We have now explained the different statistical measures in more detail. See the comment about SAL above.

Figure 5: It is difficult to have a solid conclusion here. Higher LDSA is (or should be) associated with higher number concentration. The model always fails to capture the profile in the morning hours. What are the individual contributions of MET and PSD?

Fig. 5 illustrates the vertical profiles of LDSA at the supersite. The main conclusion is that the vertical profile is very sensitive to the wind direction.

The individual contribution of MET and PSD is shown in Fig. 11. The influence of background PSD is mainly seen above the building height ($z > 30$ m), while within the street canyon, wind speed and direction strongly modify the vertical dispersion.

Are you referring to the subsection 4.2.4 Aerosol chemical composition? The conclusion has been emphasised in Section 7:

"In general, the chemical composition is acceptably reproduced except for $NH^+_4$, which is highly overestimated at all times. Yet, the performance is not always systematic with the horizontal and vertical distributions." (P26 L22).

Reading the phrase now afterwards, we totally agree. The phrase has been modified as follows: "*The difference in the modelled PSD between $M_{MET}M_{PSD}$ and $M_{MET}O_{PSD}$ is smaller than when the wind speed and/or direction are different ($O_{WD,mast}O_{PSD}$ and $O_{WD,SMEAR}O_{PSD}$, Fig. S12 in the Supplement).*" --> "The wind speed and/or direction influence the modelled PSD more than the background PSD (see Fig. S12 in the Supplement)." (P23 L14-16)

Figure 9: It might be that the coagulation (aerosol dynamics) is not fast enough. Is this a reason why fine particles are overestimated? This can be tested by aerosol dynamics on/off.

Actually, the smallest particles are underestimated at the background location. Instead, the concentration of the smallest particles are correctly simulated above the streets with traffic.

As shown in Kurppa et al. (2019), coagulation influences mainly sub-10 nm particles and their concentrations are reduced by 10% or less in a street canyon.

Figure 10 and Table S2: SAL might be a better method to compare these plots.

Thank you for the comment. We still think that SAL might not be the best methods for air quality modelling within street canyons.

The phrase has been removed when modifying the discussion section.

[revised manuscript text omitted]